# RORL: Robust Offline Reinforcement Learning via Conservative Smoothing

**Rui Yang**[1]***, Chenjia Bai**[2]*****, Xiaoteng Ma**[3]**, Zhaoran Wang**[4]**, Chongjie Zhang**[3]**, Lei Han**[5]†

[1]Hong Kong University of Science and Technology, [2]Shanghai AI Laboratory
[3]Tsinghua University, [4]Northwestern University, [5]Tencent Robotics X
ryangam@connect.ust.hk, baichenjia@pjlab.org.cn
ma-xt17@mails.tsinghua.edu.cn, zhaoranwang@gmail.com
chongjie@tsinghua.edu.cn, leihan.cs@gmail.com

## Abstract

Offline reinforcement learning (RL) provides a promising direction to exploit massive amount of offline data for complex decision-making tasks. Due to the distribution shift issue, current offline RL algorithms are generally designed to be conservative in value estimation and action selection. However, such conservatism can impair the robustness of learned policies when encountering observation deviation under realistic conditions, such as sensor errors and adversarial attacks. To trade off robustness and conservatism, we propose Robust Offline Reinforcement Learning (RORL) with a novel conservative smoothing technique. In RORL, we explicitly introduce regularization on the policy and the value function for states near the dataset, as well as additional conservative value estimation on these states. Theoretically, we show RORL enjoys a tighter suboptimality bound than recent theoretical results in linear MDPs. We demonstrate that RORL can achieve state-of-the-art performance on the general offline RL benchmark and is considerably robust to adversarial observation perturbations.

## 1 Introduction

Over the past few years, deep reinforcement learning (RL) has been a vital tool for various decision-making tasks [36, 49, 47, 11] in a trial-and-error manner. A major limitation of current deep RL algorithms is that they require intense online interactions with the environment [30, 67]. These data collecting processes can be costly and even prohibitive in many real-world scenarios such as robotics and health care [30, 53]. Offline RL [14, 28] is gaining more attention recently since it offers probabilities to learn reinforced decision-making strategies from fully offline datasets.

The main challenge of offline RL is the distribution shift between the offline dataset and the learned policy, which would lead to severe overestimation for the out-of-distribution (OOD) actions [14, 28]. To overcome such an issue, a series of model-free offline RL works [59, 14, 69, 29, 32, 2, 66, 6] propose to celebrate conservatism, such as constraining the learned policy close to the supported distribution or penalizing the $Q$-values of OOD actions. Besides, another stream of works builds upon model-based algorithms [72, 71, 58], which leverages the ensemble dynamics models to enforce pessimism through uncertainty penalizing or data generation.

However, conservatism is not the only concern when applying offline RL to the real world. Due to the sensor errors and model mismatch, the robustness of offline RL is also crucial under the realistic engineering conditions, which has not been well studied yet. In online RL, a series of works has been

---

*Equal Contribution

†Corresponding Author

36th Conference on Neural Information Processing Systems (NeurIPS 2022).

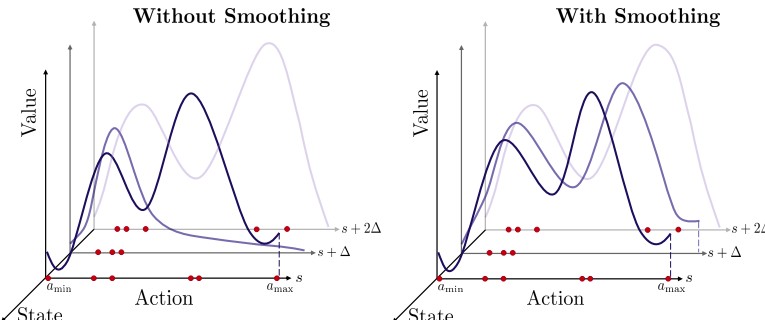

Figure 1: A schematic diagram of smoothing in offline RL. The red spots represent the offline data samples. Without state smoothing, the value function would change drastically over neighboring states and induce an unstable policy. Yet, the smoothness may also lead to value overestimation of dangerous areas. RORL trades off smoothness and possible overestimation as discussed in Sec 4.

studied to learn the optimal policy under worst-case perturbations of the observation [73, 41, 20] or environmental dynamics [57, 43, 44, 4]. Yet, it is non-trivial to apply online robust RL techniques into the offline problems. The main challenge is that the perturbation of states may bring OOD observation and extra overestimation for the value function. New techniques are needed to tackle the conservatism and robustness simultaneously in the offline RL.

This paper studies robust offline RL against adversarial observation perturbations, where the agent needs to learn the policy conservatively while handling the potential OOD observation with perturbation. We first demonstrate that current value-based offline RL algorithms lack the necessary smoothness for the policy, which is visualized in Figure 1. As an illustration, we show that a famous baseline method CQL [29] learns a non-smooth value function, leading to significant performance degradation for even a tiny scale perturbation on observation (see Section 3 for details). In addition, simply adopting the smoothing technique for existing methods may result in extra overestimation at the boundary of supported distribution and lead the agent toward unsafe areas.

To this end, we propose Robust Offline Reinforcement Learning (RORL) with a novel conservative smoothing technique, which explicitly handles the overestimation of OOD state-action pairs. Specifically, we explicitly introduce smooth regularization on both the value functions and policies for states near the dataset support and conservatively estimate the values of these OOD states based on pessimistic bootstrapping. Furthermore, we theoretically prove that RORL yields a valid uncertainty quantifier in linear MDPs and enjoys a tighter suboptimality bound than previous work [6].

In our experiments [3], we demonstrate that RORL can achieve state-of-the-art (SOTA) performance in the D4RL benchmark [12] with fewer ensemble $Q$ networks than the current SOTA approach [2]. The results of the benchmark experiments imply that robust training can lead to performance improvement in non-perturbed environments. Meanwhile, compared with current ensemble-based baselines, RORL is considerably more robust to adversarial perturbations on observations. We conduct the adversarial experiments under different attack types, showing consistently superior performance on several continuous control tasks.

## 2 Preliminaries

**Offline RL**    Considering an episodic MDP $\mathcal{M} = (\mathcal{S}, \mathcal{A}, T, r, \gamma, \mathbb{P})$, where $\mathcal{S}$ is the state space, $\mathcal{A}$ is the action space, $T$ is the length of an episode, $r$ is the reward function, $\mathbb{P}$ is the dynamics, and $\gamma$ is the discount factor. In offline RL, the objective of the agent is to find an optimal policy by sampling experiences from a fixed dataset $\mathcal{D} = \{(s_t^i, a_t^i, r_t^i, s_{t+1}^i)\}$. Nevertheless, directly applying off-policy algorithms in offline RL suffers from the distribution shift problem. In $Q$-learning, the value function evaluated on the greedy action $a'$ in Bellman operator $\mathcal{T}Q = r + \gamma \mathbb{E}_{s'}[\max_{a'}(s', a')]$ tends to have extrapolation error since $(s', a')$ has barely occurred in $\mathcal{D}$.

Pessimistic Bootstrapping for Offline RL (PBRL) [6] is an uncertainty-based method that uses bootstrapped $Q$-functions for uncertainty quantification [52] and OOD sampling for regularization.

---
[3]Our code is available at https://github.com/YangRui2015/RORL

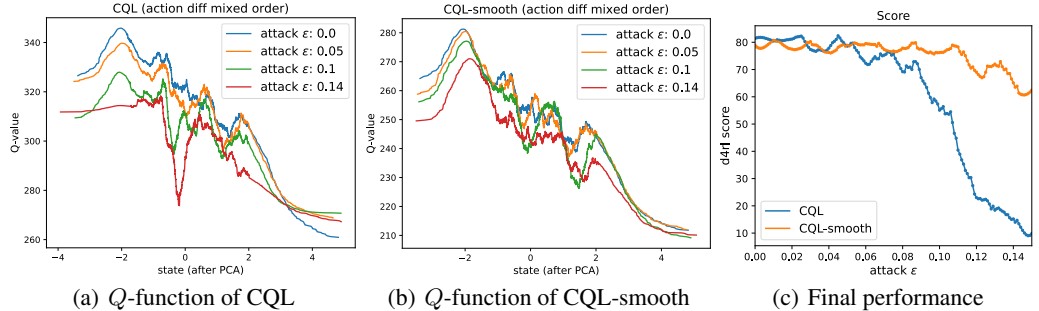

|  (a) $Q$-function of CQL | (b) $Q$-function of CQL-smooth | (c) Final performance |

Figure 2: (a) (b) The $Q$-functions of $\hat{s}$ with adversarial noises in CQL and CQL-smooth, respectively. The same moving average factor is used in plotting both figures. (c) The performance of CQL and CQL-smooth with different perturbation scales. We use 100 uniformly distributed $\epsilon \in [0.0, 0.15]$ for the evaluation.

Specifically, PBRL maintains $K$ bootstrapped $Q$ functions to quantify the epistemic uncertainty [5] and performs pessimistic update to penalize $Q$ functions with large uncertainties. The uncertainty is defined as the standard deviation among bootstrapped $Q$-functions. For each bootstrapped $Q$-function, the Bellman target is defined as $\widehat{\mathcal{T}}Q(s,a) = r(s,a) + \gamma \widehat{\mathbb{E}}_{s' \sim P(\cdot|s,a), a' \sim \pi(\cdot|s')} [Q(s', a') - \lambda u(s', a')]$. Under linear MDP assumptions, this uncertainty is equivalent to the LCB penalty and is provably efficient [24]. Furthermore, PBRL incorporates OOD sampling by sampling OOD actions to form $(s, a^{\text{ood}})$ pairs, where $a^{\text{ood}}$ follows the learned policy. The detached learning target for $(s, a^{\text{ood}})$ is $\widehat{\mathcal{T}}^{\text{ood}}Q(s, a^{\text{ood}}) := Q(s, a^{\text{ood}}) - \lambda u(s, a^{\text{ood}})$, which introduces uncertainty penalization to enforce pessimistic $Q$-functions for OOD actions.

**Smooth Regularized RL**   Robust RL aims to learn a robust policy against the adversarial perturbed environment in online RL. SR$^2$L [48] enforces smoothness in both the policy and $Q$-functions. Specifically, SR$^2$L encourages the outputs of the policy and value function to not change much when injecting small perturbations to the states. For state $s$, SR$^2$L constructs a perturbation set $\mathbb{B}_d(s, \epsilon) = \{\hat{s} : d(s, \hat{s}) \leq \epsilon\}$ with a metric $d(,)$, which is chosen to be the $\ell_p$ distance, and introduces a smoothness regularizer for policy as $\mathcal{R}_s^\pi = \mathbb{E}_{s \sim \rho^\pi} \max_{\hat{s} \in \mathbb{B}_d(s,\epsilon)} \mathcal{D}(\pi(\cdot|s) \| \pi(\cdot|\hat{s}))$, where $\mathcal{D}(\cdot\|\cdot)$ is a distance metric and the $\max$ operator gives an adversarial manner to choose $\hat{s}$. Similarly, the smoothness regularizer for the value function is defined as $\mathcal{R}_s^V = \mathbb{E}_{s \sim \rho^\pi, a \sim \pi} \max_{\hat{s} \in \mathbb{B}_d(s,\epsilon)} (Q(s,a) - Q(\hat{s}, a))^2$. SR$^2$L is shown to improve robustness against both random and adversarial perturbations.

## 3   Robustness of Offline RL: A Motivating Example

We give a motivating example to illustrate the robustness of the popular CQL [29] policies. We introduce an adversarial attack on state $s$ to obtain $\hat{s} = \arg\max_{\hat{s} \in \mathbb{B}_d(s,\epsilon)} D_{\text{J}}(\pi_\theta(\cdot|s) \| \pi_\theta(\cdot|\hat{s}))$, where $\mathbb{B}_d(s, \epsilon) = \{\hat{s} : d(s, \hat{s}) \leq \epsilon\}$ is the perturbation set and the metric $d(,)$ is chosen to be the $\ell_\infty$ norm. The Jeffrey's divergence $D_{\text{J}}$ for two distributions $P$, $Q$ is defined by: $D_{\text{J}}(P\|Q) = \frac{1}{2}[D_{\text{KL}}(P\|Q) + D_{\text{KL}}(Q\|P)]$. To obtain $\hat{s}$, we take gradient assent with respect to the loss function $D_{\text{J}}(\pi_\theta(\cdot|s) \| \pi_\theta(\cdot|\hat{s}))$ and restrict the outputs to the $\mathbb{B}_d(s, \epsilon)$ set, where $\pi_\theta$ is a learned CQL policy. We remark that the the perturbation is applied on normalized observations following prior work [73].

In the *walker-medium-v2* task from D4RL [12], we use various $\epsilon$ for adversarial attack to evaluate the robustness of CQL policies. Specifically, we use $\epsilon \in \{0, 0.05, 0.1, 0.14\}$ to control the strengths of the attack, where we have $\hat{s} = s$ if $\epsilon = 0$. Given a specific $\epsilon$, we sample $N$ state-action pairs $\{(s_i, a_i)\}$ from the offline dataset, and then perform adversarial attack to obtain $\{(\hat{s}_i, a_i)\}$ and the corresponding $Q$-values $\{Q_i(\hat{s}_i, a_i)\}$, where the $Q$-function is the trained critic of CQL.

Figure 2(a) shows the relationship between $\hat{s}_i$ and the corresponding $Q_i$ with different $\epsilon$. To visualize $\hat{s}_i$, we perform PCA dimensional reduction [55] and choose one of the reduced dimensions to represent $\hat{s}_i$. More details can be found in Appendix B.3. With the increase of $\epsilon$ in the adversarial attack, the $Q$-curve has greater deviation compared to the curve with $\epsilon = 0$. The result signifies that the $Q$-function of CQL is not smooth in the state space, which makes the adversarial noises easily affect the $Q$ values. As a comparison, we apply the proposed conservative smoothing loss in CQL

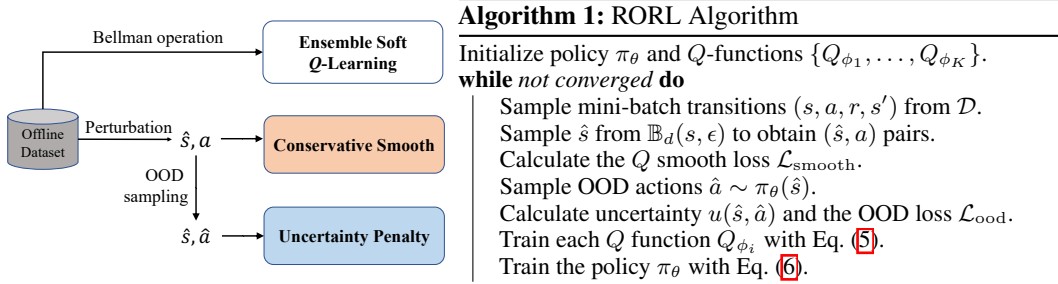

**Algorithm 1:** RORL Algorithm

Initialize policy $\pi_\theta$ and $Q$-functions $\{Q_{\phi_1}, \dots, Q_{\phi_K}\}$.
**while** *not converged* **do**
  Sample mini-batch transitions $(s, a, r, s')$ from $\mathcal{D}$.
  Sample $\hat{s}$ from $\mathbb{B}_d(s, \epsilon)$ to obtain $(\hat{s}, a)$ pairs.
  Calculate the $Q$ smooth loss $\mathcal{L}_{\text{smooth}}$.
  Sample OOD actions $\hat{a} \sim \pi_\theta(\hat{s})$.
  Calculate uncertainty $u(\hat{s}, \hat{a})$ and the OOD loss $\mathcal{L}_{\text{ood}}$.
  Train each $Q$ function $Q_{\phi_i}$ with Eq. (5).
  Train the policy $\pi_\theta$ with Eq. (6).

Figure 3: **RORL Algorithm**: RORL trains multiple $Q$-functions for uncertainty quantification. The conservative smoothing loss is calculated for $(\hat{s}, a)$ with perturbed states. We perform uncertainty penalization for $(\hat{s}, \hat{a})$ with perturbed states and OOD actions.

training (i.e., *CQL-smooth*) and use the same evaluation method to obtain $\hat{s}_i$ and $Q_i$. According to the result in Figure 2(b), the value function becomes smoother.

In addition, we show how the adversarial attack affects the final performance of offline RL policies. We use $\epsilon \in [0, 0.15]$ to evaluate both the original CQL policies (i.e., *CQL*) and CQL with conservative smoothing loss (i.e., *CQL-smooth*) in adversarial attack. Figure 2(c) shows the performance with different settings of $\epsilon$. We find that our smooth constraints significantly improve the robustness of CQL, especially for large adversarial noises.

## 4    Robust Offline RL via Conservative Smoothing

In RORL, we develop smooth regularization on both the policy and the value function for states near the dataset. The smooth constraints make the policy and the $Q$-functions robust to observation perturbations. Nevertheless, the smoothness may also lead to value overestimation in areas outside the supported dataset. To address this problem, we adopt bootstrapped $Q$-functions [39, 6] for uncertainty quantification and sample perturbed states and OOD actions for penalization. RORL obtains conservative and smooth value estimation on OOD states, which can improve the generalization ability of offline RL algorithms. The overall architecture of RORL is given in Figure 3.

**Robust $Q$-function**    We sample three sets of state-action pairs and apply different loss functions to obtain a conservative and smooth policy. Specifically, for a $(s, a)$ pair sampled from $\mathcal{D}$, we construct a perturbation set $\mathbb{B}_d(s, \epsilon)$ to obtain $(\hat{s}, a)$ pairs, where $\hat{s} \in \mathbb{B}_d(s, \epsilon)$ and $\epsilon$ is the perturbation scale. The perturbation set $\mathbb{B}_d(s, \epsilon) = \{\hat{s} : d(s, \hat{s}) \le \epsilon\}$ for state $s$ is an $\epsilon$-radius ball measured in metric $d(,)$, which is the $\ell_\infty$ norm in our paper. Then we perform OOD sampling by using the current policy $\pi_\theta$ to obtain $(\hat{s}, \hat{a})$ pairs, where $\hat{a} \sim \pi_\theta(\hat{s})$. RORL contains $K$ ensemble $Q$-functions. We denote the parameters of the $i$-th $Q$-function and the target $Q$-function as $\phi_i$ and $\phi_i'$, respectively. In the following, we give different learning targets for $(s, a)$, $(\hat{s}, a)$, and $(\hat{s}, \hat{a})$ pairs.

First, for a $(s, a)$ pair sampled from $\mathcal{D}$, we apply extended soft $Q$-learning to obtain the target as

$$\widehat{\mathcal{T}}Q_{\phi_i}(s, a) := r(s, a) + \gamma \widehat{\mathbb{E}}_{a' \sim \pi_\theta(\cdot|s')} \big[ \min_{j=1,\dots,K} Q_{\phi_j'}(s', a') - \alpha \cdot \log \pi_\theta(a'|s') \big], \qquad (1)$$

where the next-$Q$ function takes minimum value among the target $Q$-functions and $\log \pi_\theta(a'|s')$ is the entropy regularization. Note that Eq. (1) is the same learning target of SAC-$N$ in [2].

Then, for a $(\hat{s}, a)$ pair with a perturbed state, we enforce smoothness in each $Q$-function by minimizing the $Q$-value difference between $Q(s, a)$ and $Q(\hat{s}, a)$. In particular, we choose an adversarial $\hat{s} \in \mathbb{B}_d(s, \epsilon)$ that maximizes a inner objective $\mathcal{L}(Q(\hat{s}, a), Q(s, a))$, and then train each $Q$-function to minimize a loss function $\mathcal{L}_{\text{smooth}}$ with the adversarial $\hat{s}$. Intuitively, we want the $Q$-function to be smooth under the most difficult (i.e., adversarial) perturbation in $\mathbb{B}_d(s, \epsilon)$. The smooth loss function for $Q_{\phi_i}$ is as follows:

$$\mathcal{L}_{\text{smooth}}(s, a; \phi_i) = \max_{\hat{s} \in \mathbb{B}_d(s, \epsilon)} \mathcal{L}\big(Q_{\phi_i}(\hat{s}, a), Q_{\phi_i}(s, a)\big). \qquad (2)$$

We denote $\delta(s, \hat{s}, a) = Q_{\phi_i}(\hat{s}, a) - Q_{\phi_i}(s, a)$ and remark that if $\delta(s, \hat{s}, a) > 0$, the perturbed state may induce an overestimated $Q$-value that we need to smooth. In contrast, if $\delta(s, \hat{s}, a) < 0$, the

perturbed $Q$-function is underestimated, which does not cause a serious problem in offline RL. As a result, we use different weights for $\delta(s, \hat{s}, a)_+$ and $\delta(s, \hat{s}, a)_-$, where $x_+ = \max(x, 0)$ and $x_- = \min(x, 0)$. The definition of $\mathcal{L}(\cdot, \cdot)$ is give as follows:

$$\mathcal{L}\big(Q_{\phi_i}(\hat{s}, a), Q_{\phi_i}(s, a)\big) = (1 - \tau)\delta(s, \hat{s}, a)_+^2 + \tau\delta(s, \hat{s}, a)_-^2, \tag{3}$$

where we can choose $\tau \leq 0.5$. In $\mathcal{L}_{\text{smooth}}$, we does not introduce OOD action $\hat{a}$ for smoothing since the actions are desired to be close to the behavior actions for areas near the offline dataset.

Finally, to prevent overestimation of OOD states and actions, we use bootstrapped uncertainty $u(\hat{s}, \hat{a})$ as the penalty for $Q(\hat{s}, \hat{a})$, where $\hat{a} \sim \pi_\theta(\hat{s})$ is an OOD action sampled from the current policy $\pi_\theta$. We remark that a similar OOD sampling is also used in PBRL [6]. *The difference is that PBRL only penalizes the OOD actions for in-distribution states, while RORL penalizes both the OOD states and OOD actions to provide conservatism for unfamiliar areas.* We follow PBRL and use a loss function as:

$$\mathcal{L}_{\text{ood}}(s; \phi_i) = \mathbb{E}_{\hat{s}\sim\mathbb{B}_d(s,\epsilon),\hat{a}\sim\pi_\theta(\hat{s})}\big(\widehat{\mathcal{T}}_{\text{ood}}Q_{\phi_i}(\hat{s}, \hat{a}) - Q_{\phi_i}(\hat{s}, \hat{a})\big)^2, \tag{4}$$

where the pseudo-target for the OOD datapoints is computed as: $\widehat{\mathcal{T}}_{\text{ood}}Q_{\phi_i}(\hat{s}, \hat{a}) := Q_{\phi_i}(\hat{s}, \hat{a}) - u(\hat{s}, \hat{a})$, which is detached from gradients similar to the conventional TD target. The bootstrapped uncertainty $u(\hat{s}, \hat{a})$ is defined as the standard deviation among the $Q$-ensemble:

$$u(\hat{s}, \hat{a}) := \sqrt{\frac{1}{K}\sum_{k=1}^{K}\big(Q_{\phi_i}(\hat{s}, \hat{a}) - \bar{Q}_\phi(\hat{s}, \hat{a})\big)^2}.$$

The ensemble technique [39] forms an estimation of the $Q$-posterior, which yields diverse predictions and large penalty $u(\hat{s}, \hat{a})$ on areas with scarce data.

Combining the loss functions above, RORL has the following loss function for each $Q_{\phi_i}$:

$$\min_{\phi_i} \mathbb{E}_{s,a,r,s'\sim\mathcal{D}}\Big[\big(\widehat{\mathcal{T}}Q_{\phi_i}(s, a) - Q_{\phi_i}(s, a)\big)^2 + \beta_{\text{Q}}\mathcal{L}_{\text{smooth}}(s, a; \phi_i) + \beta_{\text{ood}}\mathcal{L}_{\text{ood}}(s; \phi_i)\Big], \tag{5}$$

**Robust Policy** We learn a robust policy by using a smooth constraint to make the policy change less under perturbations. Similarly, we choose an adversarial state $\hat{s} \in \mathbb{B}_d(s, \epsilon)$ that maximizes $D_{\text{J}}\big(\pi_\theta(\cdot|s)\|\pi_\theta(\cdot|\hat{s})\big)$, and then minimize the policy difference between $\pi_\theta(\cdot|s)$ and $\pi_\theta(\cdot|\hat{s})$. To conclude, we minimize the following loss function for $\pi_\theta$:

$$\min_\theta \Big[\mathbb{E}_{s\sim\mathcal{D},a\sim\pi_\theta(\cdot|s)}\big[-\min_{j=1,\ldots,K} Q_{\phi_j}(s, a) + \alpha\log\pi_\theta(a|s) + \beta_{\text{P}}\max_{\hat{s}\in\mathbb{B}_d(s,\epsilon)} D_{\text{J}}\big(\pi_\theta(\cdot|s)\|\pi_\theta(\cdot|\hat{s})\big)\big]\Big], \tag{6}$$

where the first term aims to maximize the minimum of the ensemble $Q$-functions to obtain a conservative policy, and the second term is the entropy regularization.

## 5 Theoretical Analysis

We analyze a simplified learning objective of RORL in linear MDPs [23, 24], where the feature map of the state-action pair takes the form of $\phi : \mathcal{S} \times \mathcal{A} \to \mathbb{R}^d$, and both the transition function and the reward function are assumed to be linear in $\phi$. The parameter $\widetilde{w}_t$ of RORL can be solved in closed form following the least squares value iteration (LSVI), which minimizes the following loss function.

$$\widetilde{w}_t^i = \min_{w\in\mathcal{R}^d}\Big[\sum_{i=1}^{m}\big(y_t^i - Q_w(s_t^i, a_t^i)\big)^2 + \sum_{i=1}^{m}\frac{1}{|\mathbb{B}_d(s_t^i, \epsilon)|}\sum_{\hat{s}_t^i\in\mathcal{D}_{\text{ood}}(s_t^i)}\big(Q_w(s_t^i, a_t^i) - Q_w(\hat{s}_t^i, a_t^i)\big)^2 +$$
$$\sum_{(\hat{s},\hat{a},\hat{y})\sim\mathcal{D}_{\text{ood}}}\big(\hat{y} - Q_w(\hat{s}, \hat{a})\big)^2\Big], \tag{7}$$

where we have $Q_w(s_t^i, a_t^i) = \phi(s_t^i, a_t^i)^\top w$ since the $Q$-function is also linear in $\phi$. The first term in Eq. (7) is the ordinary TD-error, where we consider the setting of $\gamma = 1$ and the $Q$-target is $y_t^i = r(s_t^i, a_t^i) + V_{t+1}(s_{t+1}^i)$. The second term is the proposed conservative smoothing loss. Specifically, $\hat{s}_t^i \sim \mathcal{D}_{\text{ood}}(s_t^i)$ are sampled from a $l_\infty$ ball of center $s_t^i$ and norm $\epsilon > 0$, which can also be formulated as $\hat{s}_t^i \sim \mathbb{B}_d(s_t^i, \epsilon)$. The third term is the additional OOD-sampling loss, which enforces

conservatism for OOD states and OOD actions. In contrast to PBRL [6], we use perturbed states sampled from $\mathcal{D}_{\text{ood}} = \bigcup_{i=1}^{m} \mathcal{D}_{\text{ood}}(s_t^i)$ rather than states from dataset. The OOD action $\hat{a}$ is sampled from policy $\pi$. The explicit solution of Eq. (7) takes the following form:

$$\widetilde{w}_t^i = \widetilde{\Lambda}_t^{-1}\Big(\sum_{i=1}^{m}\phi(s_t^i, a_t^i)y_t^i + \sum_{(\hat{s}, \hat{a}, \hat{y}) \sim \mathcal{D}_{\text{ood}}} \phi(\hat{s}, \hat{a})\hat{y}\Big), \qquad (8)$$

where the covariance matrix $\widetilde{\Lambda}_t$ is defined as

$$\begin{aligned}
\widetilde{\Lambda}_t = &\sum_{i=1}^{m}\phi(s_t^i, a_t^i)\phi(s_t^i, a_t^i)^\top + \sum_{(\hat{s}, \hat{a}) \sim \mathcal{D}_{\text{ood}}} \phi(\hat{s}_t, \hat{a}_t)\phi(\hat{s}_t, \hat{a}_t)^\top \\
&+ \sum_{i=1}^{m} \frac{1}{|\mathbb{B}_d(s_t^i, \epsilon)|} \sum_{\hat{s}_t^i \sim \mathcal{D}_{\text{ood}}(s_t^i)} \big[\phi(\hat{s}_t^i, a_t^i) - \phi(s_t^i, a_t^i)\big]\big[\phi(\hat{s}_t^i, a_t^i) - \phi(s_t^i, a_t^i)\big]^\top.
\end{aligned} \qquad (9)$$

We denote the first term and the second term as $\widetilde{\Lambda}^{\text{in}}$ and $\widetilde{\Lambda}_t^{\text{ood}}$, which represent the covariance matrices induced by the offline samples and OOD samples, respectively. Nevertheless, in linear MDPs, it is difficult to ensure the covariance $\widetilde{\Lambda}^{\text{in}} + \widetilde{\Lambda}_t^{\text{ood}} \succeq \lambda \cdot \text{I}$, since it requires that the embeddings of the samples are isotropic to make the eigenvalues of the corresponding covariance matrix lower bounded. This condition holds if we can sample embeddings uniformly from the whole embedding space. However, since the offline dataset has limited coverage in the state-action space and the OOD samples come from limited $l_\infty$-balls around the offline data, $\widetilde{\Lambda}^{\text{in}} + \widetilde{\Lambda}_t^{\text{ood}}$ cannot be guaranteed to be positive definite. PBRL [6] uses the assumption of $\widetilde{\Lambda}_t^{\text{ood}} \succeq \lambda \cdot \text{I}$, while it is unachievable empirically. In RORL, we solve this problem by introducing an additional conservative smoothing loss, which induces a covariance matrix as $\widetilde{\Lambda}_t^{\text{ood\_diff}} = \sum_{i=1}^{m} \frac{1}{|\mathbb{B}_d(s_t^i, \epsilon)|} \sum_{\hat{s}_t^i \sim \mathcal{D}_{\text{ood}}(s_t^i)}[\phi(\hat{s}_t^i, a_t^i) - \phi(s_t^i, a_t^i)][\phi(\hat{s}_t^i, a_t^i) - \phi(s_t^i, a_t^i)]^\top$ (i.e., the third term in Eq. (9)). The following theorem gives the guarantees of $\widetilde{\Lambda}_t^{\text{ood\_diff}} \succeq \lambda \cdot \text{I}$.

**Theorem 1.** *Assume $\exists i \in [1, m]$ the vector group of all $\hat{s}_t^i \sim \mathcal{D}_{ood}(s_t^i)$: $\{\phi(\hat{s}_t^i, a_t^i) - \phi(s_t^i, a_t^i)\}$ be full rank, then the covariance matrix $\widetilde{\Lambda}_t^{\text{ood\_diff}}$ is positive-definite: $\widetilde{\Lambda}_t^{\text{ood\_diff}} \succeq \lambda \cdot \text{I}$ where $\lambda > 0$.*

Recall the covariance matrix of PBRL is $\widetilde{\Lambda}_t^{\text{PBRL}} = \widetilde{\Lambda}_t^{\text{in}} + \widetilde{\Lambda}_t^{\text{ood}}$, and RORL has a covariance matrix as $\widetilde{\Lambda}_t = \widetilde{\Lambda}_t^{\text{PBRL}} + \widetilde{\Lambda}_t^{\text{ood\_diff}}$, we have the following corollary based on Theorem 1.

**Corollary 1.** *Under the linear MDP assumptions and conditions in Theorem 1, we have $\widetilde{\Lambda}_t \succeq \widetilde{\Lambda}_t^{\text{PBRL}}$. Further, the covariance matrix $\widetilde{\Lambda}_t$ of RORL is positive-definite: $\widetilde{\Lambda}_t \succeq \lambda \cdot \text{I}$, where $\lambda > 0$.*

Recent theoretical analysis shows that an appropriate uncertainty quantification is essential to provable efficiency in offline RL [24, 65, 6]. Pessimistic Value Iteration [24] defines a general $\xi$-uncertainty quantifier as the penalty and achieves provable efficient pessimism in offline RL. In linear MDPs, Lower Confidence Bound (LCB)-penalty [1, 23] is known to be a $\xi$-uncertainty quantifier for appropriately selected $\beta_t$ as $\Gamma^{\text{lcb}}(s_t, a_t) = \beta_t \cdot \big[\phi(s_t, a_t)^\top \Lambda_t^{-1}\phi(s_t, a_t)\big]^{1/2}$. Following the analysis of PBRL [6], since the bootstrapped uncertainty is an estimation of the LCB-penalty and the OOD sampling provides a covariance matrix $\widetilde{\Lambda}_t \succeq \lambda \cdot \text{I}$ given in Corollary 1, the proposed RORL also forms a valid $\xi$-uncertainty quantifier. This allows us to further characterize the optimality gap based on the pessimistic value iteration [24, 6]. We have the following suboptimality gap under linear MDP assumptions.

**Corollary 2.** $\text{SubOpt}(\pi^*, \hat{\pi}) \leq \sum_{t=1}^{T} \mathbb{E}_{\pi^*}\big[\Gamma_t^{\text{lcb}}(s_t, a_t)\big] < \sum_{t=1}^{T} \mathbb{E}_{\pi^*}\big[\Gamma_t^{\text{lcb\_PBRL}}(s_t, a_t)\big]$.

Detailed proof can be found in Appendix A. Corollary 2 indicates that RORL enjoys a tighter suboptimality bound than PBRL [6].

# 6   Experiments

We evaluate our method on the D4RL benchmark [12] with various continuous-control tasks and datasets. We compare RORL with several offline RL algorithms, including (i) BC that performs

Table 1: Normalized average returns on Gym tasks, averaged over 4 random seeds. Part of the results are reported in the EDAC paper. Top two scores for each task are highlighted.

| Task Name | BC | CQL | PBRL | SAC-10 (Reproduced) | EDAC (Paper) | EDAC-10 (Reproduced) | RORL (Ours) |
|---|---|---|---|---|---|---|---|
| halfcheetah-random | 2.2±0.0 | **31.3±3.5** | 11.0±5.8 | **29.0±1.5** | 28.4±1.0 | 13.4 ± 1.1 | 28.5±0.8 |
| halfcheetah-medium | 43.2±0.6 | 46.9±0.4 | 57.9 ±1.5 | 64.9±1.3 | **65.9±0.6** | 64.1±1.1 | **66.8±0.7** |
| halfcheetah-medium-expert | 44.0±1.6 | 95.0±1.4 | 92.3±1.1 | 107.1±2.0 | 106.3±1.9 | **107.2±1.0** | **107.8±1.1** |
| halfcheetah-medium-replay | 37.6±2.1 | 45.3±0.3 | 45.1±8.0 | **63.2±0.6** | 61.3±1.9 | 60.1±0.3 | **61.9±1.5** |
| halfcheetah-expert | 91.8±1.5 | 97.3±1.1 | 92.4±1.7 | 104.9±0.9 | **106.8±3.4** | 104.0±0.8 | **105.2±0.7** |
| hopper-random | 3.7±0.6 | 5.3±0.6 | **26.8±9.3** | 25.9±9.6 | 25.3±10.4 | 16.9±10.1 | **31.4±0.1** |
| hopper-medium | 54.1±3.8 | 61.9±6.4 | 75.3±31.2 | 0.8±0.2 | 101.6±0.6 | **103.6±0.2** | **104.8±0.1** |
| hopper-medium-expert | 53.9±4.7 | 96.9±15.1 | **110.8±0.8** | 6.1±7.7 | 110.7±0.1 | 58.1±22.3 | **112.7±0.2** |
| hopper-medium-replay | 16.6±4.8 | 86.3±7.3 | 100.6±1.0 | **102.9±0.9** | 101.0±0.5 | **102.8±0.3** | **102.8±0.5** |
| hopper-expert | 107.7±9.7 | 106.5±9.1 | **110.5±0.4** | 1.1±0.5 | 110.1±0.1 | 77.0±43.9 | **112.8±0.2** |
| walker2d-random | 1.3±0.1 | 5.4±1.7 | 8.1±4.4 | 1.5±1.1 | **16.6±7.0** | 6.7±8.8 | **21.4±0.2** |
| walker2d-medium | 70.9±11.0 | 79.5±3.2 | 89.6±0.7 | 46.7±45.3 | **92.5±0.8** | 87.6±11.0 | **102.4±1.4** |
| walker2d-medium-expert | 90.1±13.2 | 109.1±0.2 | 110.1±0.3 | **116.7±1.9** | 114.7±0.9 | 115.4±0.5 | **121.2±1.5** |
| walker2d-medium-replay | 20.3±9.8 | 76.8±10.0 | 77.7±14.5 | 89.6±3.1 | 87.1±2.3 | **94.0±1.2** | 90.4 ± 0.5 |
| walker2d-expert | 108.7±0.2 | 109.3±0.1 | 108.3±0.3 | 1.2±0.7 | **115.1±1.9** | 57.8±55.7 | **115.4 ± 0.5** |
| Average | 49.7 | 70.2 | 74.4 | 50.8 | **82.9** | 71.2 | **85.7** |
| Total | 746.1 | 1052.8 | 1116.5 | 761.6 | **1243.4** | 1068.7 | **1285.7** |

behavior cloning, (ii) CQL [29] that learns conservative value function for OOD actions, (iii) EDAC [2] that learns a diversified $Q$-ensemble to enforce conservatism, and (iv) PBRL [6] that performs uncertainty penalization and OOD sampling. We also include a basic SAC-10 algorithm as a baseline [2], which is an extension of SAC with 10 $Q$-functions. Among these methods, EDAC [2] and PBRL [6] are related to RORL since all these methods apply $Q$-ensemble for conservatism. EDAC needs much more $Q$-networks (i.e., 10~50) for hopper tasks than PBRL and RORL that only use 10 $Q$-networks. For fair comparison, we also report the reproduced results of EDAC-10. To assign uniform adversarial attack budget on each dimension of observations, we normalize the observations for SAC-10, EDAC and RORL. Besides, we use different perturbation scales for the policy smoothing loss, the Q smoothing loss and the OOD loss, namely $\epsilon_P$, $\epsilon_Q$ and $\epsilon_{ood}$. More hyper-parameters and implementation details are provided in Appendix B.

## 6.1 Benchmark Results

We evaluate each method on Gym domain that includes three environments (HalfCheetah, Hopper, and Walker2d) with five types of datasets (random, medium, medium-replay, medium-expert, and expert) for each environment. The medium-replay dataset contains experiences collected in training a medium-level policy. The random/medium/expert dataset is generated by a single random/medium/expert policy. The medium-expert dataset is a mixture of medium and expert datasets. For benchmark experiments, we set small perturbation scales $\epsilon_P$, $\epsilon_Q$, and $\epsilon_{ood}$ within $\{0.001, 0.005, 0.01\}$ when training RORL and do not include observation perturbation in the testing time.

Table 1 reports the performance of the average normalized score with standard deviation. (i) SAC-10 is unstable on several walker2d and hopper tasks since the ensemble number is relatively small to provide reliable uncertainties for SAC-$N$ [2]. (ii) EDAC solves this problem by gradient diversity constraints while still requiring 10~50 $Q$-networks to obtain reasonable performance. In contrast, RORL only uses 10 ensemble $Q$-networks to achieve better or comparable performance with EDAC. Additionally, we also show that RORL outperforms EDAC-10 by a large margin. (iii) PBRL chooses an alternative OOD-sampling technique to reduce the ensemble numbers. According to the result, RORL significantly outperforms PBRL with the same ensemble number. The reason is RORL additionally uses conservative smoothing loss for perturbed states and penalizes values of these states based on uncertainty estimation, which may improve the generalization ability of the learned policy on continuous state space. We remark that RORL significantly improves over the current SOTA results on walker2d and hopper tasks, probably because these two tasks require a more precise balance of conservatism and robustness for better performance.

## 6.2 Adversarial Attack

We adopt three attack methods, namely *random*, *action diff*, and *min Q* following prior works [73, 43]. Given perturbation scale $\epsilon$, the later two methods perform adversarial perturbation on observations

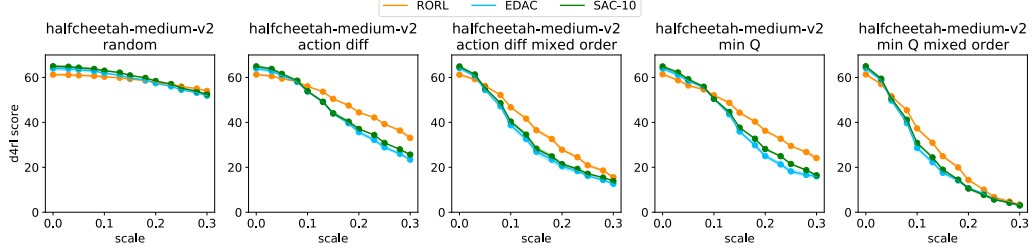

(a) Performance under attack on the halfcheetah-medium-v2 dataset

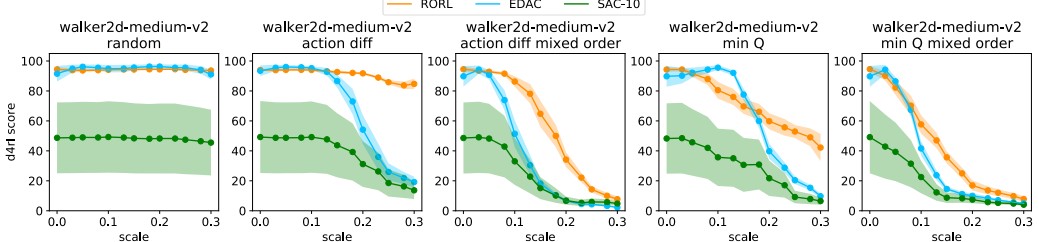

(b) Performance under attack on the walker2d-medium-v2 dataset

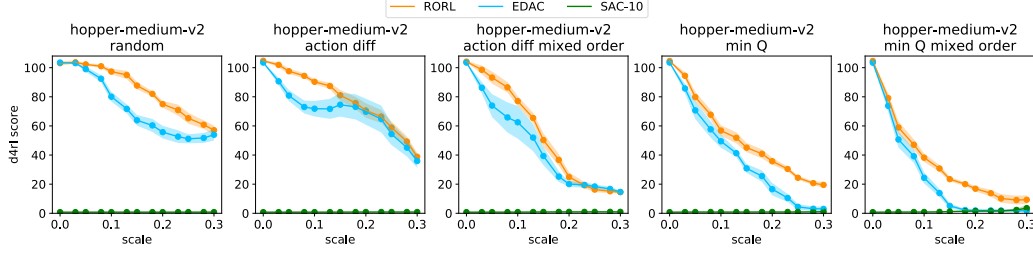

(c) Performance under attack on the hopper-medium-v2 dataset

Figure 4: (a) (b) (c) illustrate the performance of RORL, EDAC and SAC-10 under attack scales range $[0, 0.3]$ of different attack types. The curves are averaged over 4 seeds and smoothed with a window size of 3. The shaded region represents half a standard deviation.

and are given access to the agent's policy and value functions. Details about the three attack methods are as follows.

- *random* uniformly samples perturbed states in an $l_\infty$ ball of norm $\epsilon$.
- *action diff* is an effective attack based on the agent's policy and is proved to be an upper bound on the performance difference between perturbed and unperturbed environments [73]. It directly finds perturbed states in an $l_\infty$ ball of norm $\epsilon$ to satisfy: $\max_{\hat{s} \in \mathbb{B}_d(s, \epsilon)} D_J\big(\pi_\theta(\cdot|s) \| \pi_\theta(\cdot|\hat{s})\big)$, i.e., $\min_{\hat{s} \in \mathbb{B}_d(s, \epsilon)} -D_J\big(\pi_\theta(\cdot|s) \| \pi_\theta(\cdot|\hat{s})\big)$.
- *min Q* requires both the agent's policy and value function to perform a relatively stronger attack. The attacker finds a perturbed state to minimize the expected return of taking an action from that state: $\min_{\hat{s} \in \mathbb{B}_d(s, \epsilon)} Q(s, \pi_\theta(\hat{s}))$. For ensemble-based algorithms, $Q$ is set as the mean of ensemble $Q$ functions.

In our experiments, the two objectives of *action diff* and *min Q* are optimized via two ways. Specifically, we optimize the objectives through:

(1) selecting the best perturbed state from uniformly sampled 50 states, which has the advantage of simplicity and little computation cost. For attacks with this type of optimization, we use their original names without specifying.

(2) uniformly sampling 20 initial states and performing gradient decent for 10 steps with a step size of $\frac{1}{10}\epsilon$ from each initial state to find the best perturbed state. Note that we need to clip

the perturbed states within the $l_\infty$ ball at the end of each optimization step. Among the attacks using this optimization, we specifically remark "mixed-order" in their names.

We compare RORL with ensemble-based baselines EDAC and SAC-10 on halfcheetah-medium-v2, walker2d-medium-v2, and hopper-medium-v2 datasets. To handle large adversarial noise, we set the perturbation scales $\epsilon_\mathrm{P}$, $\epsilon_\mathrm{Q}$ and $\epsilon_\mathrm{ood}$ within $\{0.005, 0.02, 0.03, 0.05, 0.07\}$ in RORL's training phase. More detailed description can be found in Appendix B. The results are shown in Figure 4. In the results, RORL exhibits improved robustness than other baselines under five types of adversarial attacks. On the other hand, we find that random attack is not effective for ensemble-based offline RL algorithms, and the "mixed order" attack brings more significant performance drop than vanilla zero-order optimization.

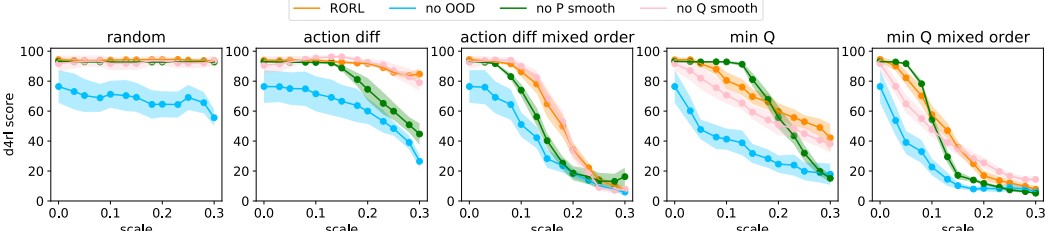

Figure 5: Ablation studies on the walker2d-medium-v2 dataset with varying perturbation scale. The curve is averaged across 4 random seeds and smoothed with a window size of 3. The shaded region represents half a standard deviation.

### 6.3 Ablations

We conduct ablation studies on the walker2d-medium-v2 dataset to evaluate the importance of three terms, i.e., the policy smoothing loss, the $Q$ smoothing term and the OOD loss. From the results in Figure 5, we can conclude that each loss contributes to the performance of RORL under adversarial observation attacks. The OOD loss is the most essential term, without which the performance is worse than RORL at almost all perturbation scales and all types of attacks. The policy smoothing loss is also important, especially for perturbation scales larger than 0.2. In addition, $Q$ smooth loss has the minimal impact, which is reasonable since the basic algorithm SAC-10 is based on 10 ensemble $Q$ networks. More ablations on the number of $Q$ networks, the effect of $\epsilon_\mathrm{ood}$ and $\tau$, and a comparison with more baselines can be found in Appendix C.

### 6.4 Computational Cost Comparison

We compare the computational cost of RORL with prior works on a single machine with one GPU (Tesla V100 32G). For each method, we measure the average epoch time (i.e., $1 \times 10^3$ training steps) and the GPU memory usage on the hopper-medium-v2 task. More discussions are provided in Appendix C.1.

As shown in Table 2, RORL runs slightly faster than CQL and much faster than PBRL. PBRL is so slow because it uses 10 $Q$ networks and needs OOD action sampling. In RORL, we also include the OOD state-action sampling

Table 2: Computational costs.

|  | Runtime (s/epoch) | GPU Memory (GB) |
|---|---|---|
| **CQL** | 32.40 | 1.4 |
| **SAC**-10 | 12.73 | 1.3 |
| **PBRL** | 102.96 | 1.8 |
| **EDAC** | 17.94 | 1.8 |
| **RORL** | 29.56 | 2.1 |

and the robust training procedure, but we implemented these procedures efficiently based on the parallelization of $Q$ networks. Even so, RORL is still slower than SAC-10 and EDAC. As demonstrated in our experiments, RORL enjoys significantly better robustness than EDAC and SAC-10 under adversarial perturbations. Regarding the GPU memory consumption, RORL uses comparable memory to PBRL and EDAC, with only 16.7% more memory usage.

## 7  Related Works

**Offline RL**   Research related to offline RL has experienced explosive growth in recent years. In model-free domain, offline RL methods focus on correcting the extrapolation error [14] in the off-

policy algorithms. The natural idea is to regularize the learned policy near the dataset distribution [59, 63, 37, 61, 69, 13, 66]. For example, MARVIL reweights the policy with exponential advantage, which implicitly guarantees the policy within the KL-divergence neighborhood of the behavior policy. Another stream of model-free methods prevents the selection of OOD actions by penalizing their $Q$-value [28, 29, 2, 10] or $V$-learning [33, 27]. With the ensemble $Q$ networks and the additional loss term to diversify their gradients, EDAC [2] achieves SOTA performance in the D4RL benchmark. Instead of diversifying gradients, PBRL [6] proposes an explicit value underestimation of OOD actions according to the uncertainty, which requires fewer ensemble networks. Inspired by EDAC and PBRL, we build our work upon ensemble networks, focusing more on the smoothness over the state space.

Besides the surprising empirical results, theoretical analysis of offline reinforcement learning algorithms is of increasing interest [9, 24, 45, 65, 70]. Though the assumptions for the dataset vary in the different papers, they all suggest that pessimism and conservatism are necessary for offline RL. Our theoretical results can be viewed as robust extensions to previous theoretical results [24, 6].

**Robust RL**  The research line of robust RL can be traced back to $H_\infty$-control theory [64, 7], where policies are optimized to be well-performed in the worst possible deterministic environment. Depending on the definition, there are different streams of research on robust RL. As the extension of robust control to MDPs, Robust MDPs (RMDPs) [38, 21, 46, 19] are proposed to formulate the perturbation of transition probabilities for MDPs. Though some recent analyses with theoretical guarantees come out under specific assumptions for RMDPs [75, 68, 31], there is currently no practical algorithm to solve RMDPs in a large-scale problem, expect some linear approximation attempt [54]. In online RL, domain randomization [56, 35] assumes the model uncertainty can be predefined in data collection by changing the setup of a simulator. However, it is not practical for offline RL. Robust Adversarial Reinforcement Learning (RARL) [43] and Noisy Robust Markov Decision Process (NR-MDP) [25] study the robust RL with the perturbed actions, showing that the policy robustness to adversarial or noisy actions can also induce robustness for model parameter changes. The most related work to ours is SR$^2$L [48], which shows policy smoothing can lead to significant performance improvement in the online setting. In contrast, we focus on the offline setting and tackle the potential overestimation of perturbed states. Another related work is S4RL [50], where the authors study different data augmentation methods to smooth observations in offline RL. Their result supports the necessity of state smoothing. More related works are discussed in Appendix E.

## 8    Conclusion

We propose Robust Offline Reinforcement Learning (RORL) to trade-off conservatism and robustness for offline RL. To achieve that, we introduce the conservative smoothing technique for the perturbed states while actively underestimating their values based on pessimistic bootstrapping to keep conservative. We show that RORL can achieve comparable or even better performance with fewer ensemble $Q$ networks than previous methods in the offline RL benchmark. In addition, we demonstrate that RORL is considerably robust to adversarial perturbations across different types of attacks. We hope our work can promote the application of offline RL under real-world engineering conditions.

The main limitation of our method is that the adversarial state sampling slows down the computing process, which may be improved in future work. Also, an interesting direction is to smooth or penalize the policy and $Q$ functions in latent spaces rather than the normalized observation space.

## Acknowledgements

This work was in part supported by Tencent Robotics X and Shanghai AI Laboratory, and in part by Science and Technology Innovation 2030 – "New Generation Artificial Intelligence" Major Project (No. 2018AAA0100904) and National Natural Science Foundation of China (62176135). The authors would like to thank the anonymous reviewers. Rui Yang thanks Yi Wang and Haoyi Song for valuable discussion.

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
