# A  Theoretical Analysis

In this section, we provide detailed theoretical analysis and proofs in linear MDPs [23].

## A.1  LSVI Solution

In linear MDPs, we assume that the transition dynamics and reward function take the form of

$$\mathbb{P}_t(s_{t+1} \,|\, s_t, a_t) = \langle \psi(s_{t+1}), \phi(s_t, a_t) \rangle, \quad r(s_t, a_t) = \theta^\top \phi(s_t, a_t), \quad \forall (s_{t+1}, a_t, s_t) \in \mathcal{S} \times \mathcal{A} \times \mathcal{S},$$
(10)

where the feature embedding $\phi : \mathcal{S} \times \mathcal{A} \mapsto \mathbb{R}^d$ is known. We further assume that the reward function $r : \mathcal{S} \times \mathcal{A} \mapsto [0, 1]$ is bounded and the feature is bounded by $\|\phi\|_2 \leq 1$.

Given the offline dataset $\mathcal{D}$, the parameter $w_t$ can be solved in the closed-form by following the LSVI algorithm, which minimizes the following loss function,

$$\widehat{w}_t = \min_{w \in \mathbb{R}^d} \sum_{i=1}^m \big(\phi(s_t^i, a_t^i)^\top w - r(s_t^i, a_t^i) - V_{t+1}(s_{t+1}^i)\big)^2$$
(11)

where $V_{t+1}$ is the estimated value function in the $(t+1)$-th step, and $y_t^i = r(s_t^i, a_t^i) + V_{t+1}(s_{t+1}^i)$ is the target of LSVI. The explicit solution to (11) takes the form of

$$\widehat{w}_t = \Lambda_t^{-1} \sum_{i=1}^m \phi(s_t^i, a_t^i) y_t^i, \quad \text{where } \Lambda_t = \sum_{i=1}^m \phi(s_t^i, a_t^i) \phi(s_t^i, a_t^i)^\top$$
(12)

## A.2  RORL Solution

In RORL, since we introduce the conservative smoothing loss and the OOD loss to learn the $Q$ value function, the parameter $\widetilde{w}_t$ of RORL can be solved as follows:

$$\widetilde{w}_t = \min_{w \in \mathcal{R}^d} \Big[ \sum_{i=1}^m \big(y_t^i - Q_w(s_t^i, a_t^i)\big)^2 + \sum_{i=1}^m \frac{1}{|\mathbb{B}_d(s_t^i, \epsilon)|} \sum_{\hat{s}_t^i \in \mathcal{D}_{\text{ood}}(s_t^i)} \big(Q_w(s_t^i, a_t^i) - Q_w(\hat{s}_t^i, a_t^i)\big)^2 +$$

$$\sum_{(\hat{s}, \hat{a}, \hat{y}) \sim \mathcal{D}_{\text{ood}}} \big(\hat{y} - Q_w(\hat{s}, \hat{a})\big)^2 \Big],$$
(13)

which is a simplified learning objective for linear MDPs. The first term is the ordinary TD-error, the second term is the $Q$ value smoothing loss, and the third term is the additional OOD loss. The explicit solution of Eq. (13) takes the following form by following LSVI:

$$\widetilde{w}_t = \widetilde{\Lambda}_t^{-1} \Big( \sum_{i=1}^m \phi(s_t^i, a_t^i) y_t^i + \sum_{(\hat{s}, \hat{a}, \hat{y}) \sim \mathcal{D}_{\text{ood}}} \phi(\hat{s}, \hat{a}) \hat{y} \Big),$$
(14)

where the covariance matrix $\widetilde{\Lambda}_t$ is defined as

$$\widetilde{\Lambda}_t = \sum_{i=1}^m \phi(s_t^i, a_t^i) \phi(s_t^i, a_t^i)^\top + \sum_{(\hat{s}, \hat{a}) \sim \mathcal{D}_{\text{ood}}} \phi(\hat{s}_t, \hat{a}_t) \phi(\hat{s}_t, \hat{a}_t)^\top$$

$$+ \sum_{i=1}^m \frac{1}{|\mathbb{B}_d(s_t^i, \epsilon)|} \sum_{\hat{s}_t^i \sim \mathcal{D}_{\text{ood}}(s_t^i)} \big[\phi(\hat{s}_t^i, a_t^i) - \phi(s_t^i, a_t^i)\big] \big[\phi(\hat{s}_t^i, a_t^i) - \phi(s_t^i, a_t^i)\big]^\top.$$
(15)

We denote the first term of Eq. (15) as $\widetilde{\Lambda}_t^{\text{in}}$, the second term as $\widetilde{\Lambda}_t^{\text{ood}}$, and the third term as $\widetilde{\Lambda}_t^{\text{ood\_diff}}$.

## A.3  $\xi$-Uncertainty Quantifier

**Theorem** (Theorem 1 restate). *Assume $\exists i \in [1, m]$ the vector group of all $\hat{s}_t^i \sim \mathcal{D}_{ood}(s_t^i)$: $\{\phi(\hat{s}_t^i, a_t^i) - \phi(s_t^i, a_t^i)\}$ is full rank, then the covariance matrix $\widetilde{\Lambda}_t^{\text{ood\_diff}}$ is positive-definite: $\widetilde{\Lambda}_t^{\text{ood\_diff}} \succeq \lambda \cdot \mathrm{I}$ where $\lambda > 0$.*

*Proof.* For the $\widetilde{\Lambda}_t^{\text{ood\_diff}}$ matrix (i.e., the third part in Eq. (15)), we denote the covariance matrix for a specific $i$ as $\Phi_t^i$. Then we have $\widetilde{\Lambda}_t^{\text{ood\_diff}} = \sum_{i=1}^m \Phi_t^i$. In the following, we discuss the condition of positive-definiteness of $\Phi_t^i$. For the simplicity of notation, we omit the superscript and subscript of $s_t^i$ and $a_t^i$ for given $i$ and $t$. Specifically, we define

$$\Phi_t^i = \frac{1}{|\mathbb{B}_d(s_t^i, \epsilon)|} \sum_{\hat{s}_j \sim \mathcal{D}_{\text{ood}}(s)} \left[\phi(\hat{s}_j, a) - \phi(s, a)\right] \left[\phi(\hat{s}_j, a) - \phi(s, a)\right]^\top,$$

where $j \in \{1, \ldots, N\}$ indicates we sample $|\mathbb{B}_d(s_t^i, \epsilon)| = N$ perturbed states for each $s$. For a nonzero vector $y \in \mathbb{R}^d$, we have

$$\begin{aligned}
y^\top \Phi_t^i y &= y^\top \left(\frac{1}{N} \sum_{j=1}^N \left(\phi(\hat{s}_j, a) - \phi(s, a)\right)\left(\phi(\hat{s}_j, a) - \phi(s, a)\right)^\top\right) y \\
&= \frac{1}{N} \sum_{j=1}^N y^\top \left(\phi(\hat{s}_j, a) - \phi(s, a)\right)\left(\phi(\hat{s}_j, a) - \phi(s, a)\right)^\top y \qquad (16) \\
&= \frac{1}{N} \sum_{j=1}^N \left(\left(\phi(\hat{s}_j, a) - \phi(s, a)\right)^\top y\right)^2 \geq 0,
\end{aligned}$$

where the last inequality follows from the observation that $\left(\phi(\hat{s}_j, a) - \phi(s, a)\right)^\top y$ is a scalar. Then $\Phi_t^i$ is always positive **semi-definite**.

In the following, we denote $z_j = \phi(\hat{s}_j, a) - \phi(s, a)$. Then we need to prove that the condition to make $\Phi_t^i$ positive **definite** is $\text{rank}[z_1, \ldots, z_N] = d$, where $d$ is the feature dimension. Our proof follows contradiction.

In Eq. (16), when $y^\top \Phi_t^i y = 0$ with a nonzero vector $y$, we have $z_j^\top y = 0$ for all $j = 1, \ldots, N$. Suppose the set $\{z_1, \ldots, z_N\}$ spans $\mathbb{R}^d$, then there exist real numbers $\{\alpha_1, \ldots, \alpha_N\}$ such that $y = \alpha_1 z_1 + \cdots + \alpha_N z_N$. But we have $y^\top y = \alpha_1 z_1^\top y + \cdots + \alpha_N z_N^\top y = \alpha_1 \times 0 + \ldots + \alpha_N \times 0 = 0$, yielding that $y = \mathbf{0}$, which forms a contradiction.

Hence, if the set $\{z_1, \ldots, z_N\}$ spans $\mathbb{R}^d$, which is equivalent to $\text{rank}[z_1, \ldots, z_N] = d$, then $\Phi_t^i$ is positive **definite**. Under the given conditions, we know that $\exists k \in [1, m]$, for any nonzero vector $y \in \mathbb{R}^d$, $y^\top \Phi_t^k y > 0$. We have $y^\top \widetilde{\Lambda}_t^{\text{ood\_diff}} y = \sum_{i=1}^m y^\top \Phi_t^i y \geq y^\top \Phi_t^k y > 0$. Therefore, $\widetilde{\Lambda}_t^{\text{ood\_diff}}$ is positive definite, which concludes our proof. $\qquad \square$

**Remark.** As a special case, when (i) the size of $\mathbb{B}_d(s_t^i, \epsilon)$ is sufficient, (ii) the dimension of states is the same as the feature $\phi(s, a)$ and $\phi(s, a) = s$ and (iii) each dimension of the state perturbation $\hat{s}_t^i - s_t^i$ is independent, the matrix $\widetilde{\Lambda}_t^{\text{ood\_diff}}$ satisfies:

$$\widetilde{\Lambda}_t^{\text{ood\_diff}} = \sum_{i=1}^m \frac{1}{|\mathbb{B}_d(s_t^i, \epsilon)|} \sum_{\hat{s}_t^i \sim \mathbb{B}_d(s_t^i, \epsilon)} (\hat{s}_t^i - s_t^i)(\hat{s}_t^i - s_t^i)^\top \approx \frac{m\epsilon^2}{3} \cdot \text{I}.$$

When we use neural networks as the feature extractor, the assumption in the above Theorem needs (i) the size of samples $\mathbb{B}_d(s_t^i, \epsilon)$ is sufficient, and (ii) the neural network maintains useful variability for state-action features. To obtain the second constraint, we require that the Jacobian matrix of $\phi(s, a)$ has full rank. Nevertheless, when we use a network as the feature embedding, such a condition can generally be met since the neural network has high randomness and nonlinearity, which results in the feature embedding with sufficient variability. Generally, we only need to enforce a bi-Lipschitz continuity for the feature embedding. We denote $x_1 = (s_1, a)$ and $x_2 = (s_2, a)$ as two different inputs. $x_1^k$ is the $k$-th dimension of $x_1$. The bi-Lipschitz constraint can be formed as

$$C_1 \|x_1^k - x_2^k\|_{\mathcal{X}} \leq \|\phi(x_1) - \phi(x_2)\|_{\Phi} \leq C_2 \|x_1^k - x_2^k\|_{\mathcal{X}}, \quad \forall k \in (1, |\mathcal{X}|), \qquad (17)$$

where $C_1 < C_2$ are two positive constants. The lower-bound $C_1$ ensures the features space has enough variability for perturbed states, and the upper-bound can be obtained by Spectral regularization

[17] that makes the network easy to coverage. An approach to obtain bi-Lipschitz continuity is to regularize the norm of the gradients by using the gradient penalty as

$$\mathcal{L}_{\text{bilip}} = \mathbb{E}_x \big[ \big( \min \big( \|\nabla_{x^k} \phi(x)\| - C_1, 0 \big) \big)^2 + \big( \max \big( \|\nabla_{x^k} \phi(x)\| - C_2, 0 \big) \big)^2 \big], \quad \forall k \in (1, |\mathcal{X}|).$$

In experiments, we do not use explicit constraints (e.g., Spectral regularization) for the upper bound since the state has relatively low dimensions, and we find a small fully connected network does not resulting in a large $C_2$ empirically.

Recall the covariance matrix of PBRL is $\widetilde{\Lambda}_t^{\text{PBRL}} = \widetilde{\Lambda}_t^{\text{in}} + \widetilde{\Lambda}_t^{\text{ood}}$, and RORL has a covariance matrix as $\widetilde{\Lambda}_t = \widetilde{\Lambda}_t^{\text{PBRL}} + \widetilde{\Lambda}_t^{\text{ood\_diff}}$, we have the following corollary based on Theorem 1.

**Corollary** (Corollary 1 restate). *Under the linear MDP assumptions and conditions in Theorem 1, we have $\widetilde{\Lambda}_t \succeq \widetilde{\Lambda}_t^{\text{PBRL}}$. Further, the covariance matrix $\widetilde{\Lambda}_t$ of RORL is positive-definite: $\widetilde{\Lambda}_t \succeq \lambda \cdot \mathrm{I}$, where $\lambda > 0$.*

Recent theoretical analysis shows that an appropriate uncertainty quantification is essential for provable efficiency in offline RL [24, 65, 6]. Pessimistic Value Iteration [24] defines a general $\xi$-uncertainty quantifier as the penalty and achieves provable efficient pessimism in offline RL. We give the definition of a $\xi$-uncertainty quantifier as follows.

**Definition 1** ($\xi$-Uncertainty Quantifier [24]). *The set of penalization $\{\Gamma_t\}_{t \in [T]}$ forms a $\xi$-Uncertainty Quantifier if it holds with probability at least $1 - \xi$ that*

$$|\widehat{\mathcal{T}} V_{t+1}(s, a) - \mathcal{T} V_{t+1}(s, a)| \le \Gamma_t(s, a)$$

*for all $(s, a) \in \mathcal{S} \times \mathcal{A}$, where $\mathcal{T}$ is the Bellman operator and $\widehat{\mathcal{T}}$ is the empirical Bellman operator that estimates $\mathcal{T}$ based on the data.*

In linear MDPs, Lower Confidence Bound (LCB)-penalty [1, 23] is known to be a $\xi$-uncertainty quantifier for appropriately selected $\beta_t$ as $\Gamma^{\text{lcb}}(s_t, a_t) = \beta_t \cdot \big[ \phi(s_t, a_t)^\top \Lambda_t^{-1} \phi(s_t, a_t) \big]^{1/2}$. Following the analysis of PBRL [6], since the bootstrapped uncertainty is an estimation of the LCB-penalty, the proposed RORL also form a valid $\xi$-uncertainty quantifier with the covariance matrix $\widetilde{\Lambda}_t \succeq \lambda \cdot \mathrm{I}$ given in Corollary 1.

**Theorem 2.** *For all the OOD datapoint $(\hat{s}, \hat{a}, \hat{y}) \in \mathcal{D}_{\text{ood}}$, if we set $\hat{y} = \mathcal{T} V_{t+1}(s^{\text{ood}}, a^{\text{ood}})$, it then holds for $\beta_t = \mathcal{O}\big( T \cdot \sqrt{d} \cdot log(T/\xi) \big)$ that*

$$\Gamma_t^{\text{lcb}}(s_t, a_t) = \beta_t \big[ \phi(s_t, a_t)^\top \widetilde{\Lambda}_t^{-1} \phi(s_t, a_t) \big]^{1/2} \tag{18}$$

*forms a valid $\xi$-uncertainty quantifier, where $\widetilde{\Lambda}_t$ is the covariance matrix of RORL.*

*Proof.* The proof follows that of the analysis of PBRL [6] in linear MDPs [24]. We define the empirical Bellman operator of RORL as $\widetilde{\mathcal{T}}$, then

$$\widetilde{\mathcal{T}} V_{t+1}(s_t, a_t) = \phi(s_t, a_t)^\top \widetilde{w}_t,$$

where $\widetilde{w}_t$ follows the solution in Eq. (14). Then it suffices to upper bound the following difference between the empirical Bellman operator and Bellman operator

$$\mathcal{T} V_{t+1}(s, a) - \widetilde{\mathcal{T}} V_{t+1}(s, a) = \phi(s, a)^\top (w_t - \widetilde{w}_t).$$

Here we define $w_t$ as follows

$$w_t = \theta + \int_{\mathcal{S}} V_{t+1}(s_{t+1}) \psi(s_{t+1}) \mathrm{d}s_{t+1}, \tag{19}$$

where $\theta$ and $\psi$ are defined in Eq. (10). It then holds that

$$\mathcal{T} V_{t+1}(s, a) - \widetilde{\mathcal{T}} V_{t+1}(s, a) = \phi(s, a)^\top (w_t - \widetilde{w}_t)$$

$$= \phi(s, a)^\top w_t - \phi(s, a)^\top \widetilde{\Lambda}_t^{-1} \sum_{i=1}^m \phi(s_t^i, a_t^i) \big( r(s_t^i, a_t^i) + V_{t+1}^i(s_{t+1}^i) \big)$$

$$- \phi(s, a)^\top \widetilde{\Lambda}_t^{-1} \sum_{(\hat{s}, \hat{a}, \hat{y}) \in \mathcal{D}_{\text{ood}}} \phi(\hat{s}, \hat{a}) \hat{y}. \tag{20}$$

where we plug the solution of $\widetilde{w}_t$ in Eq. (14). Meanwhile, by the definitions of $\widetilde{\Lambda}_t$ and $w_t$ in Eq. (15) and Eq. (19), respectively, we have

$$
\begin{aligned}
\phi(s,a)^\top w_t &= \phi(s,a)^\top \widetilde{\Lambda}_t^{-1} \widetilde{\Lambda}_t w_t \\
&= \phi(s,a)^\top \widetilde{\Lambda}_t^{-1} \bigg( \sum_{i=1}^m \phi(s_t^i, a_t^i) \mathcal{T}V_{t+1}(s_t, a_t) + \sum_{(\hat{s}, \hat{a}, \hat{y}) \in \mathcal{D}_{\text{ood}}} \phi(\hat{s}, \hat{a}) \mathcal{T}V_{t+1}(\hat{s}, \hat{a}) +
\end{aligned}
$$

$$
\sum_{i=1}^m \frac{1}{|\mathbb{B}_d(s_t^i, \epsilon)|} \sum_{\hat{s}_t^i \sim \mathcal{D}_{\text{ood}}(s_t^i)} [\phi(\hat{s}_t^i, a_t^i) - \phi(s_t^i, a_t^i)][\phi(\hat{s}_t^i, a_t^i) - \phi(s_t^i, a_t^i)]^\top w_t \bigg). \tag{21}
$$

Plugging Eq. (21) into Eq. (20) yields

$$
\mathcal{T}V_{t+1}(s,a) - \widetilde{\mathcal{T}}V_{t+1}(s,a) = \text{(i)} + \text{(ii)} + \text{(iii)}, \tag{22}
$$

where we define

$$
\text{(i)} = \phi(s,a)^\top \widetilde{\Lambda}_t^{-1} \sum_{i=1}^m \phi(s_t^i, a_t^i) \big( \mathcal{T}V_{t+1}(s_t^i, a_t^i) - r(s_t^i, a_t^i) - V_{t+1}^i(s_{t+1}^i) \big),
$$

$$
\text{(ii)} = \phi(s,a)^\top \widetilde{\Lambda}_t^{-1} \sum_{(\hat{s}, \hat{a}, \hat{y}) \in \mathcal{D}_{\text{ood}}} \phi(\hat{s}, \hat{a}) \big( \mathcal{T}V_{t+1}(\hat{s}, \hat{a}) - \hat{y} \big),
$$

$$
\text{(iii)} = \phi(s,a)^\top \widetilde{\Lambda}_t^{-1} \sum_{i=1}^m \frac{1}{|\mathbb{B}_d(s_t^i, \epsilon)|} \sum_{\hat{s}_t^i \sim \mathcal{D}_{\text{ood}}(s_t^i)} \bigg[ \Big( \phi(\hat{s}_t^i, a_t^i) \phi(\hat{s}_t^i, a_t^i)^\top w_t - \phi(\hat{s}_t^i, a_t^i) \phi(s_t^i, a_t^i)^\top w_t \Big)
$$

$$
+ \Big( \phi(s_t^i, a_t^i) \phi(s_t^i, a_t^i)^\top w_t - \phi(s_t^i, a_t^i) \phi(\hat{s}_t^i, a_t^i)^\top w_t \Big) \bigg].
$$

Following the standard analysis based on the concentration of self-normalized process [1, 3, 60, 23, 24] and the fact that $\Lambda_{\text{ood}} \succeq \lambda \cdot I$, it holds that

$$
|\text{(i)}| \leq \beta_t \cdot \big[ \phi(s_t, a_t)^\top \Lambda_t^{-1} \phi(s_t, a_t) \big]^{1/2}, \tag{23}
$$

with probability at least $1 - \xi$, where $\beta_t = \mathcal{O}\big( T \cdot \sqrt{d} \cdot \log(T/\xi) \big)$. Meanwhile, by setting $y = \mathcal{T}V_{t+1}(s^{\text{ood}}, a^{\text{ood}})$, it holds that (ii) $= 0$. For (iii), we have

$$
\Big( \phi(\hat{s}_t^i, a_t^i) \phi(\hat{s}_t^i, a_t^i)^\top w_t - \phi(\hat{s}_t^i, a_t^i) \phi(s_t^i, a_t^i)^\top w_t \Big) + \Big( \phi(s_t^i, a_t^i) \phi(s_t^i, a_t^i)^\top w_t - \phi(s_t^i, a_t^i) \phi(\hat{s}_t^i, a_t^i)^\top w_t \Big)
$$

$$
= \phi(\hat{s}_t^i, a_t^i) \Big( \mathcal{T}V_{t+1}(\hat{s}_t^i, a_t^i) - \mathcal{T}V_{t+1}(s_t^i, a_t^i) \Big) + \phi(s_t^i, a_t^i) \Big( \mathcal{T}V_{t+1}(s_t^i, a_t^i) - \mathcal{T}V_{t+1}(\hat{s}_t^i, a_t^i) \Big)
$$

$$
= \big( \phi(\hat{s}_t^i, a_t^i) - \phi(s_t^i, a_t^i) \big) \big( \mathcal{T}V_{t+1}(\hat{s}_t^i, a_t^i) - \mathcal{T}V_{t+1}(s_t^i, a_t^i) \big) \tag{24}
$$

Since we enforce smoothness for the value function, we have $\mathcal{T}V_{t+1}(\hat{s}_t^i, a_t^i) \approx \mathcal{T}V_{t+1}(s_t^i, a_t^i)$. Thus (iii) $\approx 0$. To conclude, we obtain from Eq. (22) that

$$
|\mathcal{T}V_{t+1}(s,a) - \widetilde{\mathcal{T}}V_{t+1}(s,a)| \leq \beta_t \cdot \big[ \phi(s_t, a_t)^\top \Lambda_t^{-1} \phi(s_t, a_t) \big]^{1/2} \tag{25}
$$

for all $(s,a) \in \mathcal{S} \times \mathcal{A}$ with probability at least $1 - \xi$. $\qquad\square$

## A.4 Suboptimality Gap

Theorem 2 allows us to further characterize the optimality gap based on the pessimistic value iteration [24]. First, we give the following lemma.

**Lemma 1.** *Given two positive definite matrix A and B, it holds that:*

$$
\frac{x^\top A^{-1} x}{x^\top (A+B)^{-1} x} > 1. \tag{26}
$$

*Proof.* Leveraging the properties of generalized Rayleigh quotient, we have

$$
\frac{x^\top A^{-1} x}{x^\top (A+B)^{-1} x} \geq \lambda_{\min}\big( (A+B) A^{-1} \big) = \lambda_{\min}\big( I + B A^{-1} \big) = 1 + \lambda_{\min}\big( B A^{-1} \big). \tag{27}
$$

Since $B$ and $A^{-1}$ are both positive definite, the eigenvalues of $BA^{-1}$ are all positive: $\lambda_{\min}\big( BA^{-1} \big) > 0$. This ends the proof. $\qquad\square$

Then, according to the definition of LCB-penalty in Eq. (18), since $\widetilde{\Lambda}_t = \widetilde{\Lambda}_t^{\text{PBRL}} + \widetilde{\Lambda}_t^{\text{ood\_diff}}$ with $\widetilde{\Lambda}_t^{\text{ood\_diff}} \succeq \lambda\mathrm{I}$. we have the relationship of the LCB-penalty between RORL and PBRL as follows.

**Corollary 3.** *Suppose $\Lambda_t^{PBRL}$ is positive definite. The RORL-induced LCB-penalty term is less than the PBRL-induced LCB-penalty, as $\Gamma_t^{\text{lcb}}(s_t, a_t) = \beta_t\big[\phi(s_t, a_t)^\top \widetilde{\Lambda}_t^{-1}\phi(s_t, a_t)\big]^{1/2} < \Gamma_t^{\text{lcb\_PBRL}}(s_t, a_t)$.*

*Proof.* Since $\widetilde{\Lambda}_t = \widetilde{\Lambda}_t^{\text{PBRL}} + \widetilde{\Lambda}_t^{\text{ood\_diff}}$ and $\widetilde{\Lambda}_t^{\text{ood\_diff}} \succeq \lambda I$, we have

$$\frac{\phi(s_t, a_t)^\top \widetilde{\Lambda}_t^{-1}\phi(s_t, a_t)}{\phi(s_t, a_t)^\top (\widetilde{\Lambda}_t^{\text{PBRL}})^{-1}\phi(s_t, a_t)} = \frac{\phi(s_t, a_t)^\top (\widetilde{\Lambda}_t^{\text{PBRL}} + \Lambda_t^{\text{ood\_diff}})^{-1}\phi(s_t, a_t)}{\phi(s_t, a_t)^\top (\widetilde{\Lambda}_t^{\text{PBRL}})^{-1}\phi(s_t, a_t)} < 1. \tag{28}$$

where the inequality directly follows Lemma 1. Then we have

$$\phi(s_t, a_t)^\top \widetilde{\Lambda}_t^{-1}\phi(s_t, a_t) < \phi(s_t, a_t)^\top (\widetilde{\Lambda}_t^{\text{PBRL}})^{-1}\phi(s_t, a_t). \tag{29}$$

$\square$

Theorem 2 and Corollary 3 allow us to further characterize the optimality gap of the pessimistic value iteration. In particular, we have the following suboptimality gap under linear MDP assumptions.

**Corollary** (Corollary 2 restate). *Under the same conditions as Theorem 2, it holds that* $\text{SubOpt}(\pi^*, \hat{\pi}) \leq \sum_{t=1}^T \mathbb{E}_{\pi^*}\big[\Gamma_t^{\text{lcb}}(s_t, a_t)\big] < \sum_{t=1}^T \mathbb{E}_{\pi^*}\big[\Gamma_t^{\text{lcb\_PBRL}}(s_t, a_t)\big].$

We refer to Jin et al [24] for a detailed proof of the first inequality. The second inequality is directly induced by $\Gamma_t^{\text{lcb}}(s_t, a_t) < \Gamma_t^{\text{lcb\_PBRL}}(s_t, a_t)$ in Corollary 3. The optimality gap is information-theoretically optimal under the linear MDP setup with finite horizon [24]. Therefore, RORL enjoys a tighter suboptimality bound than PBRL [6] in linear MDPs.

## B  Implementation Details and Experimental Settings

In this section, we provide detailed implementation and experimental settings.

### B.1  Implementation Details

**SAC-10**  Our SAC-10 implementation is based on [2], which is open-source. We keep the default parameters as EDAC [2] except for the ensemble size set to 10 in our paper. In addition, we normalize each dimension of observations to a standard normal distribution for consistency with RORL. The hyper-parameters are listed in Table 3.

Table 3: Hyper-parameters of SAC-10

| Hyper-parameters | Value |
|---|---|
| The number of bootstrapped networks $K$ | 10 |
| Policy network | FC(256,256,256) with ReLU activations |
| $Q$-network | FC(256,256,256) with ReLU activations |
| Target network smoothing coefficient $\tau$ for every training step | 5e-3 |
| Discount factor $\gamma$ | 0.99 |
| Policy learning rate | 3e-4 |
| $Q$ network learning rate | 3e-4 |
| Optimizer | Adam |
| Automatic Entropy Tuning | True |
| batch size | 256 |

**EDAC**  Our EDAC implementation is based on the open-source code of the original paper [2]. In the benchmark results, we directly report results from the paper which are the previous SOTA performance on the D4RL Mujoco benchmark. As for other experiments, we also normalize the observations and use 10 ensemble $Q$ networks for consistency with RORL, and set the gradient diversity term $\eta = 1$ by default.

**RORL** We implement RORL based on SAC-10 and keep the hyper-parameters the same. The differences are the introduced policy and $Q$ network smoothing techniques and the additional value underestimation on OOD state-action pairs. In Eq. (5), the coefficient $\beta_Q$ for the $Q$ network smoothing loss $\mathcal{L}_{\text{smooth}}$ is set to 0.0001 for all tasks, and the coefficient $\beta_{\text{ood}}$ for the OOD loss $\mathcal{L}_{\text{ood}}$ is tuned within $\{0.0, 0.1, 0.5\}$. Besides, the coefficient $\beta_P$ of the policy smoothing loss in Eq. (6) is searched in $\{0.1, 1.0\}$. When training the policy and value functions in RORL, we randomly sample $n$ perturbed observations from a $l_\infty$ ball of norm $\epsilon$ and select the one that maximizes $D_J(\pi_\theta(\cdot|s)\|\pi_\theta(\cdot|\hat{s}))$ or $\mathcal{L}_{\text{smooth}}$, respectively. We denote the perturbation scales for the $Q$ value functions, the policy, and the OOD loss as $\epsilon_Q$, $\epsilon_P$ and $\epsilon_{\text{ood}}$. The number of sampled perturbed observations $n$ is tuned within $\{10, 20\}$. The OOD loss underestimates the values for $n$ perturbed states $\hat{s} \sim \mathbb{B}_d(s, \epsilon)$ with actions sampled from the current policy $\hat{a} \sim \pi_\theta(\hat{s})$. For each $\hat{s}$, we sample a single $\hat{a}$ for the OOD loss. Regarding the $Q$ smoothing loss in Eq. (3), the parameter $\tau$ is set to 0.2 in all tasks for conservative value estimation. All the hyper-parameters used in RORL for the benchmark experiments and adversarial experiments are listed in Table 4 and Table 5 respectively. Note that for halfcheetah tasks, 10 ensemble $Q$ networks already enforce sufficient pessimism for OOD state-action pairs, thus we do not need additional OOD loss for these tasks.

As for the OOD loss $\mathcal{L}_{\text{ood}}$ in Eq. (4), we remark that the pseudo-target $\widehat{\mathcal{T}}_{\text{ood}} Q_{\phi_i}(\hat{s}, \hat{a})$ for the OOD state-action pairs $(\hat{s}, \hat{a})$ can be implemented in two ways: $\widehat{\mathcal{T}}_{\text{ood}} Q_{\phi_i}(\hat{s}, \hat{a}) := Q_{\phi_i}(\hat{s}, \hat{a}) - \lambda u(\hat{s}, \hat{a})$ and $\widehat{\mathcal{T}}_{\text{ood}} Q_{\phi_i}(\hat{s}, \hat{a}) := \min_{i=1,\dots,K} Q_{\phi_i}(\hat{s}, \hat{a})$. We refer to the two targets as the "minus target" and the "min target", and compare them in Appendix C.14. Intuitively, the "minus target" introduces an additional parameter $\lambda$ but is more flexible to tune for different environments and different types of data. In contrast, the "min target" requires tuning the number of ensemble $Q$ networks and cannot enforce appropriate conservatism for all tasks given only 10 ensemble $Q$ networks. Following PBRL [6], we also decay the OOD regularization coefficient $\lambda$ with decay pace $d$ for each training step to stabilize $\mathcal{L}_{\text{ood}}$, because we need strong OOD regularization at the beginning of training and need to avoid too large OOD loss that leads the value function to be fully negative. $\lambda$ and $d$ are also listed in the two tables.

Table 4: Hyper-parameters of RORL for the benchmark results

| Task Name | $\beta_Q$ | $\beta_P$ | $\beta_{\text{ood}}$ | $\epsilon_Q$ | $\epsilon_P$ | $\epsilon_{\text{ood}}$ | $\tau$ | $n$ | $\lambda\,(d)$ |
|---|---|---|---|---|---|---|---|---|---|
| halfcheetah-random
halfcheetah-medium
halfcheetah-medium-expert
halfcheetah-medium-replay
halfcheetah-expert | 0.0001 | 0.1 | 0.0 | 0.001
0.001
0.001
0.001
0.005 | 0.001
0.001
0.001
0.001
0.005 | 0.00 | 0.2 | 20
10
10
10
10 | 0 |
| hopper-random
hopper-medium
hopper-medium-expert
hopper-medium-replay
hopper-expert | 0.0001 | 0.1 | 0.5 | 0.005 | 0.005 | 0.01 | 0.2 | 20 | $1 \to 0.5\ (1e^{-6})$
$2 \to 0.1\ (1e^{-6})$
$3 \to 1.0\ (1e^{-6})$
$0.1 \to 0\ (1e^{-6})$
$4 \to 1\ (1e^{-6})$ |
| walker2d-random
walker2d-medium
walker2d-medium-expert
walker2d-medium-replay
walker2d-expert | 0.0001 | 1.0 | 0.5
0.1
0.1
0.1
0.5 | 0.005
0.01
0.01
0.01
0.005 | 0.005
0.01
0.01
0.01
0.005 | 0.01 | 0.2 | 20 | $5.0 \to 0.5\ (1e^{-5})$
$1 \to 0.1\ (5e^{-7})$
$0.1 \to 0.1\ (0.0)$
$0.1 \to 0.1\ (0.0)$
$1.0 \to 0.5\ (1e^{-6})$ |

Table 5: Hyper-parameters of RORL for the adversarial attack results

| Task Name | $\beta_Q$ | $\beta_P$ | $\beta_{\text{ood}}$ | $\epsilon_Q$ | $\epsilon_P$ | $\epsilon_{\text{ood}}$ | $\tau$ | $n$ | $\lambda\,(d)$ |
|---|---|---|---|---|---|---|---|---|---|
| halfcheetah-medium
walker2d-medium
hopper-medium | 0.0001 | 1
0.5
0.1 | 0.0
0.5
0.5 | 0.03
0.03
0.005 | 0.05
0.07
0.005 | 0.00
0.03
0.02 | 0.2 | 20 | 0
$1 \to 0.1\ (1e^{-6})$
$2 \to 0.1\ (1e^{-6})$ |

## B.2 Experimental Settings

For all experiments, we train algorithms for 3000 epochs (1000 training steps per epoch, i.e., 3 million steps in total) following EDAC [2]. We use small perturbation scales to train the $Q$ networks and the policy network for the benchmark experiments and relatively large scales for the adversarial attack experiments as listed in Table 4 and Table 5.

In the benchmark results, we evaluate algorithms for 1000 steps in clean environments (without adversarial attack) at the end of each epoch. The reported results are normalized to d4rl scores that measure how the performance compared with the expert score and the random score: thenormalized score $= 100 \times \frac{\text{score}-\text{random score}}{\text{expert score}-\text{random score}}$. Besides, the benchmark results are averaged over 4 random seeds. Regarding the adversarial attack experiments, we evaluate algorithms in perturbed environments that performing "random", "action diff", and "min Q" attack with zeroth-order and mixed-order optimizations as discussed in Section 6.2. Similar to prior work [73], agents receive observations with malicious noise and the environments do not change their internal transition dynamics. We evaluate each algorithm for 10 trajectories (1000 steps per trajectory) and average their returns over 4 random seeds.

## B.3 Visualization Settings of CQL

For visualizing the relationship between the $Q$-function and the state space (i.e., Figure 2 and Figure 6), we sample 2560 adversarial transitions from the offline dataset for each attack $\epsilon$ and calculate the corresponding $Q$-function. Since the state has relatively high dimensions (i.e., 11 or 17), we perform PCA dimensional reduction to reduce the state to 4 dimensions. We find the $Q$-function generally has a strong correlation to one or two dimensions of the state after dimensional reduction. For other dimensions, the relationship between the $Q$-value and the PCA-reduced state often has one or two peaks, which has less variety in the curve.

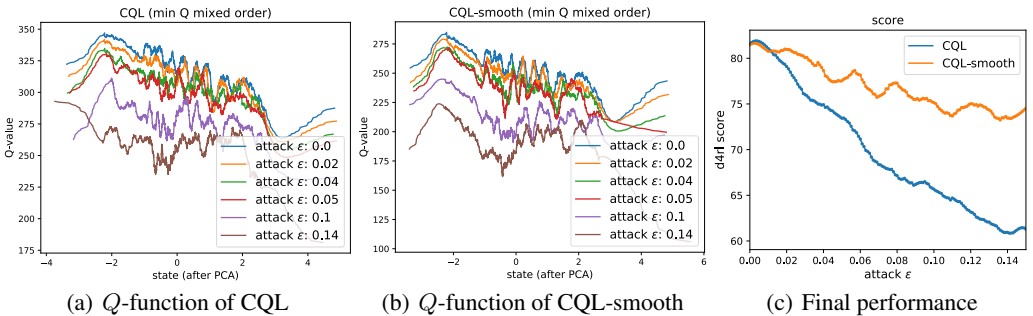

(a) $Q$-function of CQL     (b) $Q$-function of CQL-smooth     (c) Final performance

Figure 6: (a)(b) The $Q$-functions of $\hat{s}$ with 'min $Q$ mixed order' adversarial noises in CQL and CQL-smooth, respectively. The same moving average factor is used in plotting both figures. (c) The performance evaluation of CQL and CQL-smooth with different perturbation scales. We use 100 different $\epsilon \in [0.0, 0.15]$ for the evaluation.

# C Additional Experimental Results

In this section, we present additional ablation studies and adversarial experiments.

## C.1 Computational Cost Comparison

In this subsection, we compare the computational cost of RORL with prior works on a single machine with one GPU (Tesla V100 32G) and one CPU (Intel Xeon Platinum 8255C @ 2.50GHz). For each method, we measure the average epoch time (i.e., $1 \times 10^3$ training steps) and the GPU memory usage on the hopper-medium-v2 task. For CQL, PBRL, SAC-$N$, and EDAC, we evaluate the computational cost based on their official code.

As shown in Table 2, RORL runs slightly faster than CQL, mainly because CQL needs the OOD action sampling and the logsumexp approximation. For ensemble-based baselines, RORL runs

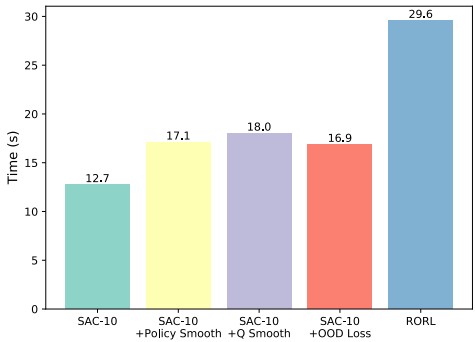
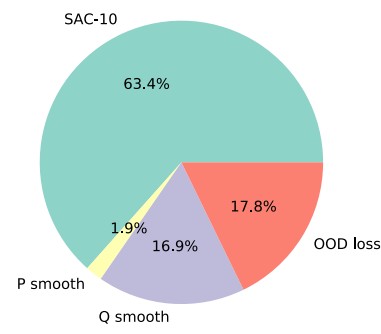

(a) Average epoch time of RORL's components     (b) Memory usage of RORL's components

Figure 7: Visualization of the average epoch time and memory usage for RORL and its components.

much faster than PBRL, requiring only 28.7% of PBRL's epoch time. PBRL is so slow because it uses 10 ensemble $Q$ networks for uncertainty measure and needs OOD action sampling for value underestimation. In RORL, we also include the OOD state-action sampling and additional adversarial training procedures, but we implement these procedures efficiently based on GPU operation and parallelism. Even so, RORL is still slower than SAC-10 and EDAC. But as demonstrated in our experiments, RORL enjoys significantly better robustness than EDAC and SAC-10 under different types of perturbations. As for the GPU memory consumption, RORL uses comparable memory to PBRL and EDAC, with only 16.7% more memory usage.

Furthermore, we analyze the computational cost of RORL's components ($Q$ smoothing, policy smoothing, and the OOD loss). Specifically, we measure the average epoch time of *SAC-10+Policy Smooth*, *SAC-10+Q Smooth*, *SAC-10+OOD Loss* in Figure 7(a), and calculate the corresponding memory usage of each component in Figure 7(b). For the training time, *SAC-10+Q Smooth* runs the slowest and *SAC-10+Policy Smooth* runs slightly slower than *SAC-10+OOD Loss*. This is mainly because sampling the worst-case perturbation occupies the most time. In addition, since we use an ensemble of 10 $Q$ networks, the memory usage of the $Q$ smoothing loss and the OOD loss (both need to pass $n$ perturbed states to 10 $Q$ networks) is larger than the policy smoothing loss.

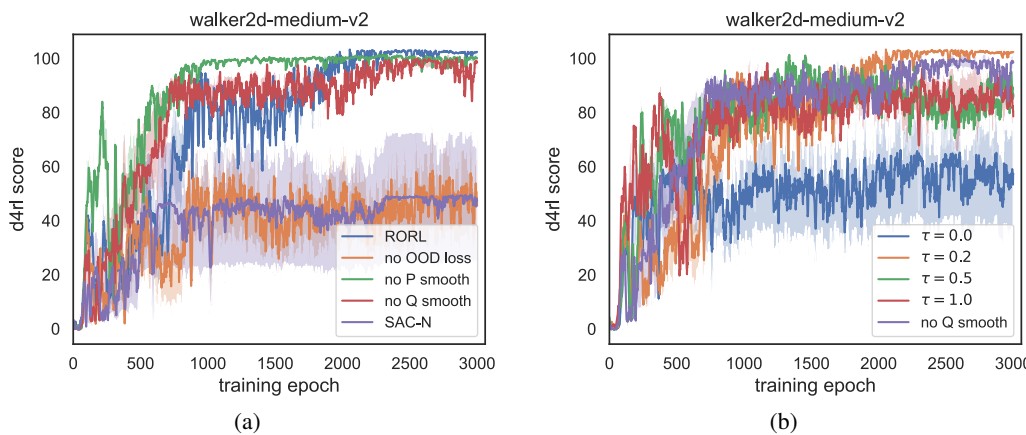

(a)                                      (b)

Figure 8: (a) Ablation studies of three introduced loss. The "P smooth" and the "$Q$ smooth" refer to the policy smoothing loss and the $Q$ network smoothing loss. (b) Ablations studies of the hyper-parameter $\tau$ in the benchmark experiments.

## C.2 Ablations on Benchmark Results

In the benchmark experiments, RORL outperforms other baselines, especially in walker2d tasks. We conduct ablation studies on this task to verify the effectiveness of RORL's components. In Figure 8

(a), we can find that each introduced loss (i.e., the OOD loss, the policy smoothing loss and the $Q$ smoothing loss) influences the performance on the walker2d-medium-v2 task. Specifically, the OOD loss affects the most, without which the performance would drop close to SAC-N's performance. In addition, the $Q$ smoothing loss is helpful for stabilizing the training and final performance in clean environments.

In Figure 8 (b), we evaluate the performance of RORL with varying $\tau$. The results suggest that $\tau$ is an important factor that balances the learning of in-distribution and out-of-distribution $Q$ values. In Eq. (3), we want to assign larger weights $(1 - \tau)$ on the $\delta(s, \hat{s}, a)^2_+$ and smaller weights $(\tau)$ on the $\delta(s, \hat{s}, a)^2_-$ to underestimate the values of OOD states, where $\delta(s, \hat{s}, a) = Q_{\phi_i}(\hat{s}, a) - Q_{\phi_i}(s, a)$. On the contrary, a too small $\tau$ can also lead to overestimation of in-distribution state-action pairs. In Figure 8 (b), $\tau = 0$ leads to poor performance while larger $\tau = 0.5, 1.0$ also result in performance worse than RORL without $Q$ smoothing. Empirically, we find $\tau = 0.2$ works well across different tasks and set $\tau = 0.2$ by default for all experiments.

In the above analysis, we know that the OOD loss is a key component in RORL. We further study the impact of the OOD loss and $\epsilon_{\text{ood}}$ on the performance and the value estimation. As shown in Figure 9 (a), when $\epsilon_{\text{ood}} = 0$, the performance of RORL drops significantly, which illustrates the effectiveness of underestimating values of OOD states since the smoothness of RORL may overestimate these values. From Figure 9 (b), we can verify that the OOD loss with $\epsilon_{\text{ood}} > 0$ contributes to the value underestimation.

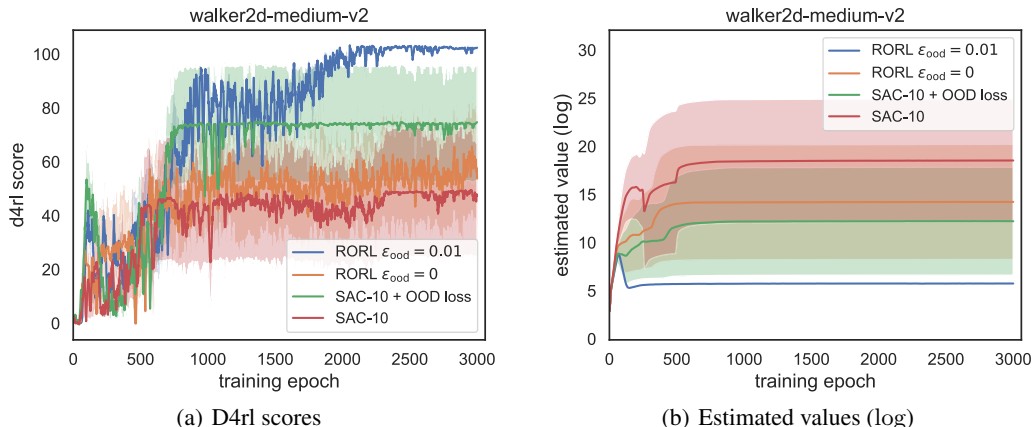

(a) D4rl scores        (b) Estimated values (log)

Figure 9: The ablations of the OOD loss $\mathcal{L}_{\text{ood}}$ and the hyper-parameter $\epsilon_{\text{ood}}$ on the benchmark experiments.

### C.3 Robustness Measures

In prior works [48, 73], the authors only demonstrate the robustness of algorithms via comparing the return curves with different attack scales. To better measure the robustness of RL algorithms, we consider the *robust score* as the areas under the perturbation curve in Figure 4. Since the returns in the figure have been normalized as introduced in Appendix B.2, we can simply calculate the *robust score* for each attack strategy as:

$$\text{robust score} = \frac{1}{N} \sum_{i \in [1,N]} Rs[i]$$

where $Rs$ is the list of returns under $N$ monotonically increasing attack scales. The introduced *robust score* treats different attack scales equally. However, in many real scenarios, we would pay more attention to larger-scale disturbances. To this end, we also define a *weighted robust score* as:

$$\text{weighted robust score} = \frac{2}{(1 + N) \times N} \sum_{i \in [1,N]} i \times Rs[i]$$

where the weights are assigned according to the scale order. In Table 6 and Table 7, RORL consistently outperforms EDAC and SAC-10 on the two robustness metrics. For walker2d and hopper tasks,

Table 6: Robust scores under attack on halfcheetah-medium-v2, walker2d-medium-v2, and hopper-medium-v2 tasks.

| Task | | Random | Action Diff | Action Diff Mixed Order | Min Q | Min $Q$ Mixed Order | Average |
|------|------|--------|-------------|-------------------------|-------|---------------------|---------|
| halfcheetah-m | RORL | 58.6 | 49.5 | 38.0 | 43.5 | 28.2 | 43.6 |
| | EDAC | 59.2 | 44.5 | 33.0 | 38.1 | 25.0 | 40.0 |
| | SAC-10 | 60.1 | 45.6 | 34.2 | 39.8 | 25.7 | 41.1 |
| walker2d-m | RORL | 94.1 | 91.0 | 56.9 | 71.0 | 43.3 | 71.2 |
| | EDAC | 95.1 | 68.3 | 37.2 | 62.1 | 35.9 | 59.7 |
| | SAC-10 | 48.2 | 37.0 | 23.0 | 29.2 | 18.5 | 31.2 |
| hopper-m | RORL | 84.8 | 78.4 | 53.9 | 51.5 | 34.7 | 60.7 |
| | EDAC | 72.2 | 69.7 | 45.5 | 38.3 | 23.7 | 49.9 |
| | SAC-10 | 0.79 | 0.82 | 0.89 | 0.88 | 1.36 | 0.95 |

Table 7: Weighted robust scores under attack on halfcheetah-medium-v2, walker2d-medium-v2, and hopper-medium-v2 tasks.

| Task | | Random | Action Diff | Action Diff Mixed Order | Min Q | Min $Q$ Mixed Order | Average |
|------|------|--------|-------------|-------------------------|-------|---------------------|---------|
| halfcheetah-m | RORL | 57.4 | 44.5 | 29.7 | 37.0 | 17.7 | 37.2 |
| | EDAC | 57.0 | 37.0 | 23.9 | 28.7 | 14.4 | 32.2 |
| | SAC-10 | 57.9 | 38.3 | 25.1 | 30.8 | 14.9 | 33.4 |
| walker2d-m | RORL | 94.1 | 89.1 | 39.1 | 61.8 | 26.7 | 62.2 |
| | EDAC | 95.1 | 52.9 | 18.7 | 45.7 | 18.8 | 46.2 |
| | SAC-10 | 47.7 | 30.1 | 13.8 | 21.3 | 10.7 | 24.7 |
| hopper-m | RORL | 76.0 | 68.0 | 36.1 | 37.4 | 21.1 | 47.7 |
| | EDAC | 61.7 | 61.4 | 30.8 | 21.7 | 9.6 | 37.0 |
| | SAC-10 | 0.80 | 0.84 | 0.93 | 0.91 | 1.67 | 1.03 |

RORL surpasses EDAC by more than 10 points on both the *robust score* and the *weighted robust score*.

### C.4 Ablations of Components in the Adversarial Experiments

In Section 6.3, we conducted ablations of RORL's major components in the adversarial settings. In this subsection, we provide robust scores of the ablation results over 4 random seeds in Table 8. Besides, results of $\epsilon_{\mathrm{ood}} = 0$ are also included to demonstrate the effectiveness of penalizing values of OOD states. From Table 8, we can conclude that the OOD loss is the most essential component of RORL, and only penalizing in-distribution states is insufficient for adversarial perturbations. To summarize, the order of the importance of each component is: OOD loss $> \epsilon_{\mathrm{ood}} >$ policy smoothing loss $> Q$ smoothing loss. The conclusion may be different for different tasks, for example we found that the halfcheetah task does not even need the OOD loss because the SAC-10 framework already provides it with sufficient pessimism.

### C.5 Ablations on the Number of Ensemble $Q$ Networks

We conduct the adversarial attack experiments with different number of bootstrapped $Q$ networks in RORL. As shown in Figure 10, the robustness of RORL improves as the ensemble size $K$ increases. For $K = 6, 8, 10$, RORL has similar initial performance but $K = 10$ considerably outperforms others as the attack scale increases. Therefore, we set $K = 10$ by default in our paper.

### C.6 Ablations of $\tau$ for the Adversarial Experiments

In this subsection, we study the performance under attacks with varying $\tau \in \{0.0, 0.2, 0.5, 1.0\}$. From the results in Figure 11, we find $\tau = 0.2$ slightly outperforms the others on 4 out of the 5

Table 8: The robust scores of ablation studies on the walker2d-medium-v2 task

| | random | action diff | action diff mixed order | min $Q$ | min $Q$ mixed order | Average Score |
|---|---|---|---|---|---|---|
| RORL | 94.1 | 90.9 | 56.9 | 71.0 | 43.3 | 71.2 |
| no OOD | 68.4 | 62.0 | 37.6 | 35.9 | 22.0 | 45.2 |
| no P smooth | 92.8 | 78.7 | 48.6 | 67.1 | 39.3 | 65.3 |
| no $Q$ smooth | 92.7 | 91.1 | 57.3 | 62.2 | 40.2 | 68.7 |
| $\epsilon_{ood}$=0 | 74.1 | 70.5 | 44.1 | 46.3 | 26.9 | 52.4 |

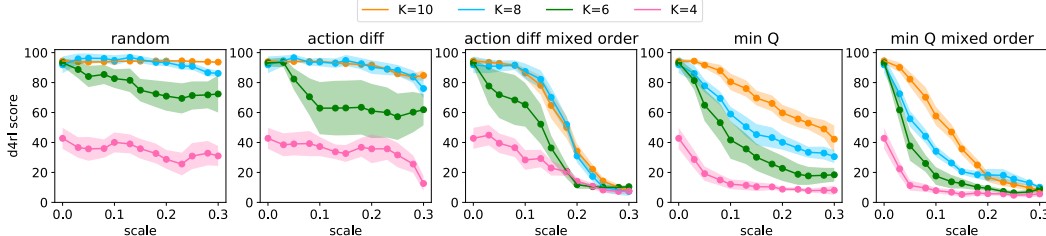

Figure 10: Ablations on the number of $Q$ networks on the walker2d-medium-v2 dataset.

attack types. The results are also consistent with the ablation studies of the benchmark experiments in Appendix C.2. Accordingly, we set $\tau = 0.2$ by default for all experiments in our paper.

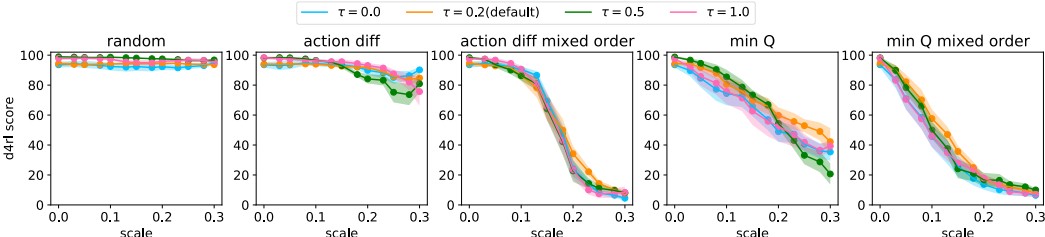

Figure 11: Comparison of different $\tau$ in the adversarial experiments on the walker2d-medium-v2 dataset.

## C.7 Ablations on the Number of Sampled Perturbed Observations

We ablate the number of sampled perturbed observations in Figure 12. From the figure, we can conclude that the robustness of RORL improves as the number of samples $n$ increases. At the same time, the computational cost also increases as $n$ increases. Therefore, we can choose $n$ according to the computational budget. Interestingly, RORL with $n = 1$ already outperforms SAC-10 by a large margin, which could be an appropriate option when computing resources are limited.

## C.8 Adversarial Attack with Different $Q$ Functions

In our experiments, it is assumed that the 'min $Q$' and the 'min $Q$ mixed order' attackers have access to the corresponding $Q$ value functions of the attacked agent. Generally, the assumption is strong for many real-world scenarios. In addition, the comparison does not take into account the impact of attacking with different $Q$ functions. Intuitively, conservative and smoothed $Q$ functions make it easier for attackers to find the most impactful perturbation to degrade the performance. To investigate the impact of different $Q$ functions, we swap the attacker's $Q$-function, i.e. **using RORL's $Q$-functions to attack EDAC and using EDAC's $Q$-functions to attack RORL**. In Figure 13, we can conclude:

(1) RORL outperforms EDAC with a wider margin when using the same $Q$ functions. Surprisingly, the difference of normalized scores increases from 37.3 to 51.2 for walker2d-medium-v2 task with the largest 'min Q' attack.

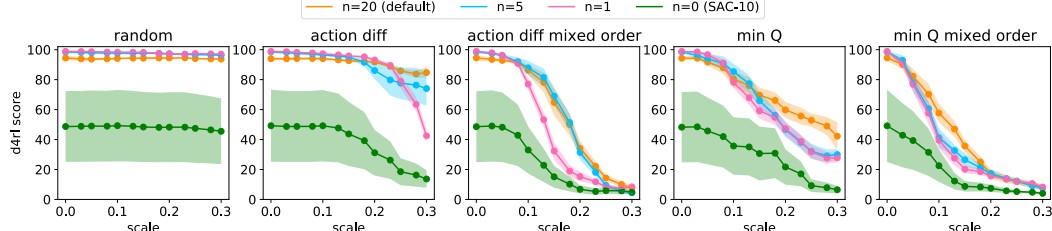

Figure 12: Ablations on the number of sampled perturbed observations. The comparison is made on the walker2d-medium-v2 task.

(2) The value function of EDAC may still not be smooth and can mislead the attackers. In contrast, RORL successfully learns smooth value functions, which may facilitate further research on stronger attack strategies for robust offline RL.

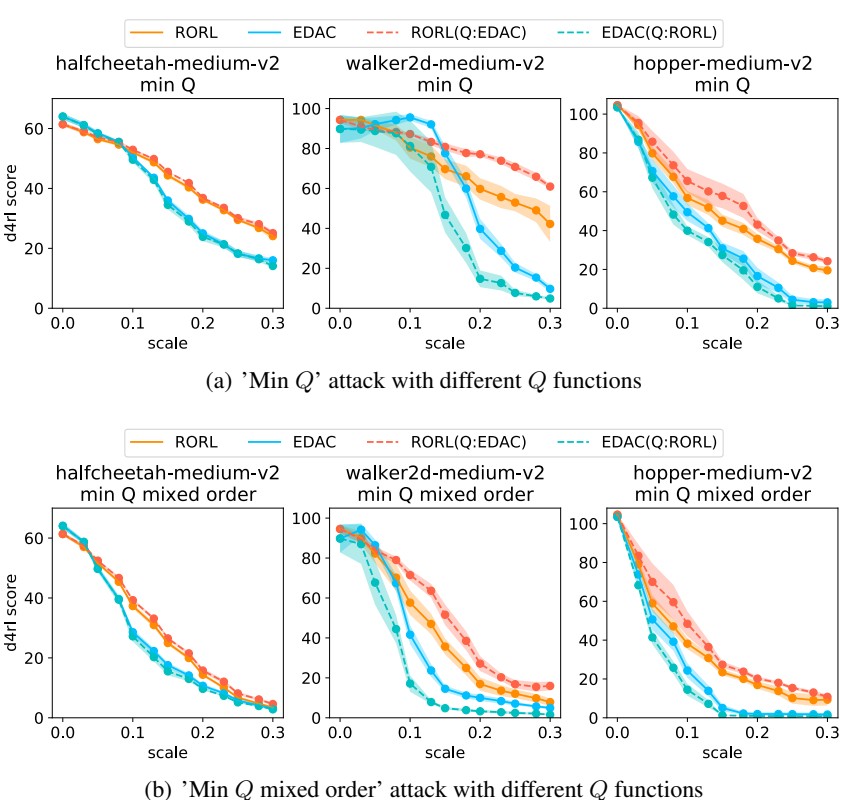

(a) 'Min $Q$' attack with different $Q$ functions

(b) 'Min $Q$ mixed order' attack with different $Q$ functions

Figure 13: Performance under the 'min $Q$' and the 'min $Q$ mixed order' adversarial attacks with different $Q$ functions. Curves are averaged over 4 random seeds. *RORL(Q:EDAC)* refers to attacking RORL with EDAC's $Q$ functions, and *EDAC(Q:RORL)* refers to attacking EDAC with RORL's $Q$ functions. When the attacker uses the same $Q$ functions, RORL outperforms EDAC with a wider margin.

## C.9   Comparison with EDAC+Smoothing

We also compare EDAC with both policy smoothing and $Q$ smoothing, which leverages the gradient penalty rather than our OOD loss to enforce pessimism on OOD state-action pairs. The hyper-parameters are kept the same with EDAC and RORL, except $\tau = 0.5$ in EDAC+Smoothing. As shown in Figure 14, the smoothing technique slightly improves the robustness of EDAC under large-scale (0.2∼0.3) adversarial perturbations, but it significantly decreases the overall performance under

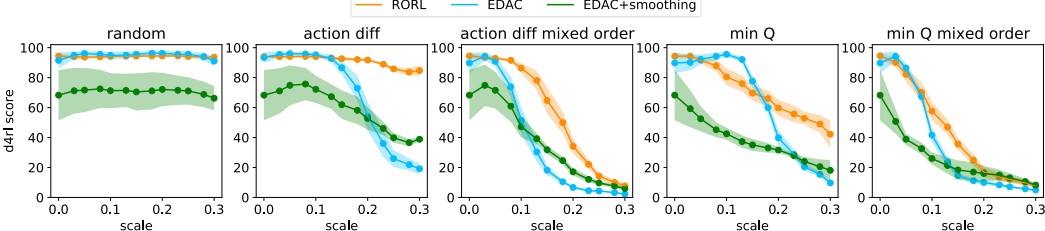

Figure 14: Comparison with EDAC+Smoothing under adversarial attacks on the walker2d-medium-v2 task. The curves are averaged over 4 seeds and smoothed with a window size of 3.

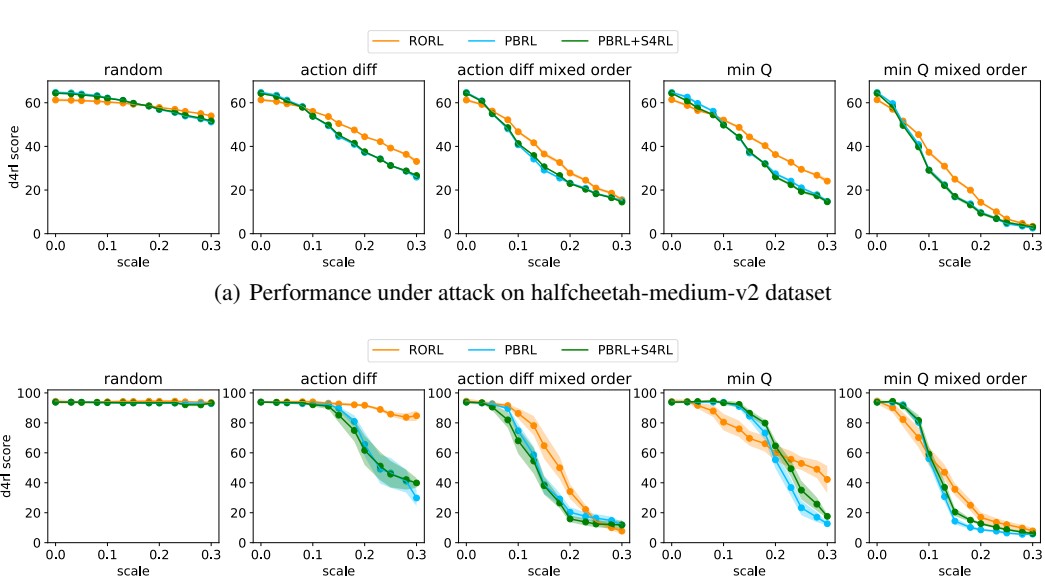

(a) Performance under attack on halfcheetah-medium-v2 dataset

(b) Performance under attack on walker2d-medium-v2 dataset

Figure 15: Comparison of PBRL and PBRL+S4RL under attack scales range [0, 0.3] of different types of attack. The curves are averaged over 4 seeds and smoothed with a window size of 3. The shaded region represents half a standard deviation.

attack. The results imply that directly using smoothing techniques without explicit OOD penalization can even worsen the robust scores of previous SOTA offline RL algorithm.

### C.10 Comparison with PBRL + S4RL

We also include comparison with PBRL and PBRL+S4RL to verify if RORL is more robust than data augmentation for offline RL [50]. The main differences between RORL and S4RL are three folds:

(1) S4RL only implicitly smooths the value functions while RORL explicitly smooths them, which is more efficient and enjoys theoretical guarantees.

(2) S4RL does not consider the impact of overestimation on OOD states brought by the data augmentation, which can be harmful for offline RL. In contrast, RORL further underestimates values for OOD states, which essentially alleviates the potential overestimation.

(3) In addition, S4RL selects adversarially perturbed states according to the gradient of $Q(s, \pi(s))$, aiming to choose the direction where the $Q$-value deviates the most. Different from S4RL, RORL samples perturbed states to maximize a conservative smoothing loss $\mathcal{L}\big(Q_{\phi_i}(\hat{s}, a), Q_{\phi_i}(s, a)\big)$ and a policy smoothing loss $\max_{\hat{s} \in \mathbb{B}_d(s, \epsilon)} D_{\mathrm{J}}\big(\pi_\theta(\cdot|s) \| \pi_\theta(\cdot|\hat{s})\big)$ defined in Section 4.

The empirical results on halfcheetah-medium-v2 and walker2d-medium-v2 are shown in Figure 15. We can observe that S4RL only slightly improves the robustness of PBRL on the walker2d-medium-

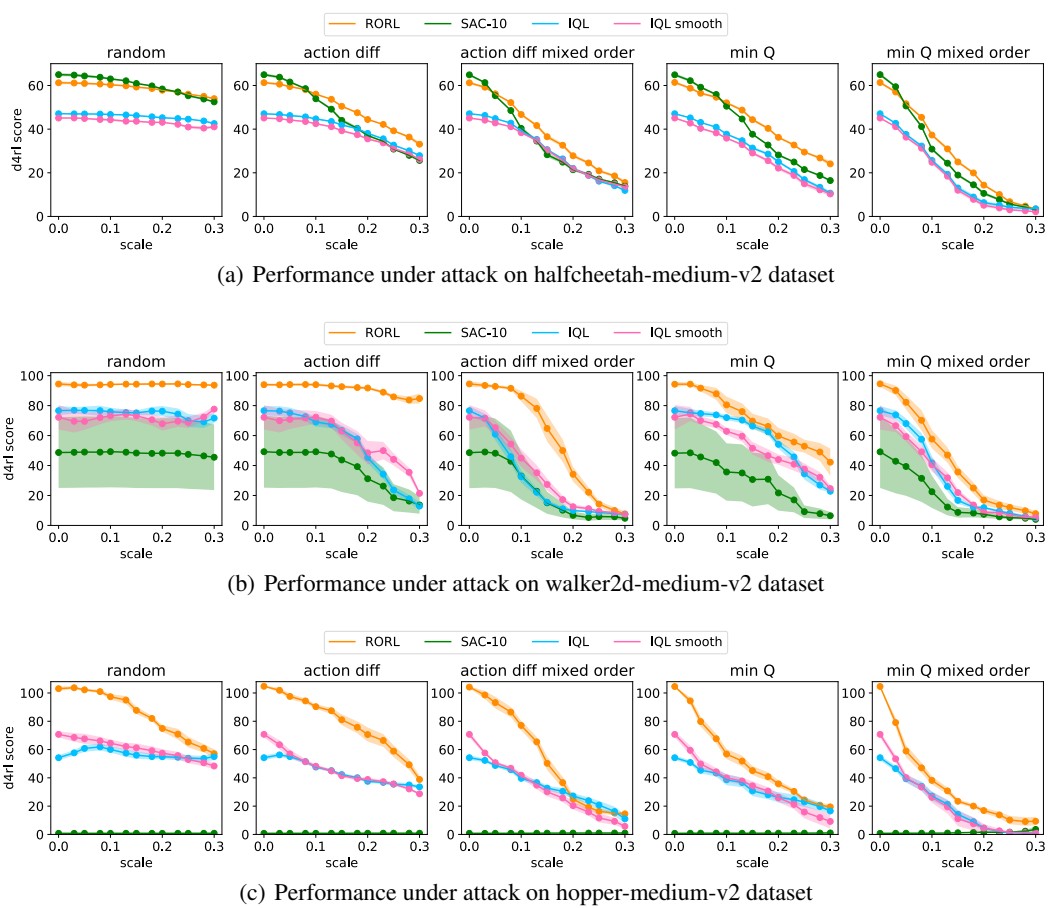

(a) Performance under attack on halfcheetah-medium-v2 dataset

(b) Performance under attack on walker2d-medium-v2 dataset

(c) Performance under attack on hopper-medium-v2 dataset

Figure 16: Comparison of IQL and IQL smooth. Figures (a) (b) (c) illustrate the performance under attack scales range $[0, 0.3]$ of different types of attack. The curves are averaged over 4 seeds and smoothed with a window size of 3. The shaded region represents half a standard deviation.

v2 task and has little impact on the halfcheetah-medium-v2 task. In contrast, RORL exhibits higher robustness across different tasks and attack types.

### C.11  Combining Smoothing with IQL

We combine the policy smoothing and $Q$ function smoothing techniques in RORL with IQL [27], a SOTA offline RL algorithm without ensemble $Q$ networks. We use the default hyper-parameters of IQL and set the hyper-parameters for smoothing the same as in Table 5. The training and evaluation settings keep the same as the adversarial experiments in our paper. As shown in Figure 16, we can observe that IQL with the smoothing technique (short for 'IQL smooth') slightly improves the robustness on the walker2d-medium-v2 and hopper-medium-v2 tasks, but it has little effect on the halfcheetah-medium-v2 task. This suggests that simply adopting the smoothing technique does not consistently improve the performance in the offline setting. In contrast, RORL introduces additional OOD underestimation based on uncertainty measure, which helps to obtain conservatively smoothed policy and value functions.

### C.12  Comparing the 'max' and the 'mean' Operators in Smoothing

In our implementation, we first sample $n$ perturbed states and select the one that maximizes the smoothing losses in Eq. (2) and Eq. (6). It is interesting to see if the 'max' operator is useful, as we can also use the 'mean' operator as an alternative, i.e., $\mathcal{L}^{mean}_{\text{smooth}}(s, a; \phi_i) = \mathbb{E}_{\hat{s} \in \mathbb{B}_d(s,\epsilon)} \mathcal{L}\big(Q_{\phi_i}(\hat{s}, a), Q_{\phi_i}(s, a)\big)$ and $\mathbb{E}_{\hat{s} \in \mathbb{B}_d(s,\epsilon)} D_J\big(\pi_\theta(\cdot|s) \| \pi_\theta(\cdot|\hat{s})\big)$.

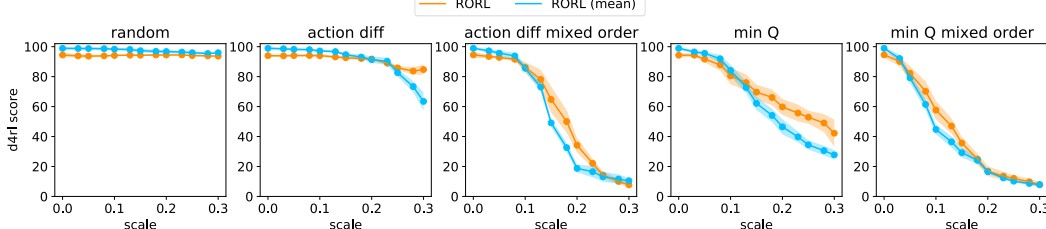

Figure 17: Comparing the 'max' with the 'mean' operators in our smoothing techniques. The comparison is made on the walker2d-medium-v2 task.

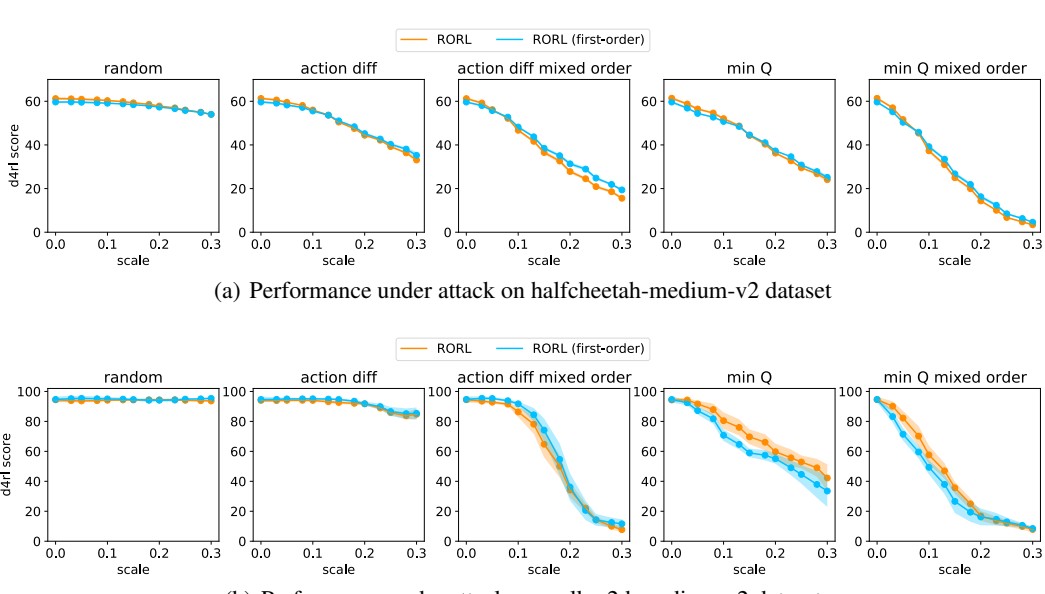

(a) Performance under attack on halfcheetah-medium-v2 dataset

(b) Performance under attack on walker2d-medium-v2 dataset

Figure 18: Comparison of zeroth-order and first-order optimization in the training period. The curves are averaged over 4 seeds and smoothed with a window size of 3. The shaded region represents half a standard deviation.

The results are demonstrated in Figure 17. We can find that RORL with the 'max' operator obtains a more conservative policy under small-scale perturbations and achieves higher robustness under large-scale perturbations. Since the 'max' operator has the same complexity as the 'mean' operator, we use the 'max' operator by default, which is also a zeroth-order approximation to an inner optimization problem.

### C.13 Comparing Different Optimization for Perturbation Generation during Training

In the training period, we use zeroth-order optimization to approximately optimize the $Q$ smoothing loss in Eq. (2) and the policy smoothing loss: $\max_{\hat{s} \in \mathbb{B}_d(s, \epsilon)} D_J\big(\pi_\theta(\cdot|s) \| \pi_\theta(\cdot|\hat{s})\big)$. In this way, we can accelerate training the robust policy and obtain similar performance. Besides, zeroth-order optimization is commonly applied in black-box attack where we can only access the input and output of neural networks without explicit gradient information. Black-box attack for reinforcement learning might be a promising direction in the future.

We also implemented a first-order version of RORL, which requires an average epoch time of 72.7s on a V100 GPU (while the average epoch time of the zeroth-order method is 29.6s). Since the perturbation generation for each training step is independent, we use the first-order optimization for a probability of 0.5 to alleviate the computational cost. In Figure 18, we compare the trained policies with zeroth-order and first-order optimization. We can conclude that the two types of optimization for perturbation generation have very similar performance. On halfcheetah-medium task, the first-order version performs slightly better than the zeroth-order version, while the zeroth-order version works

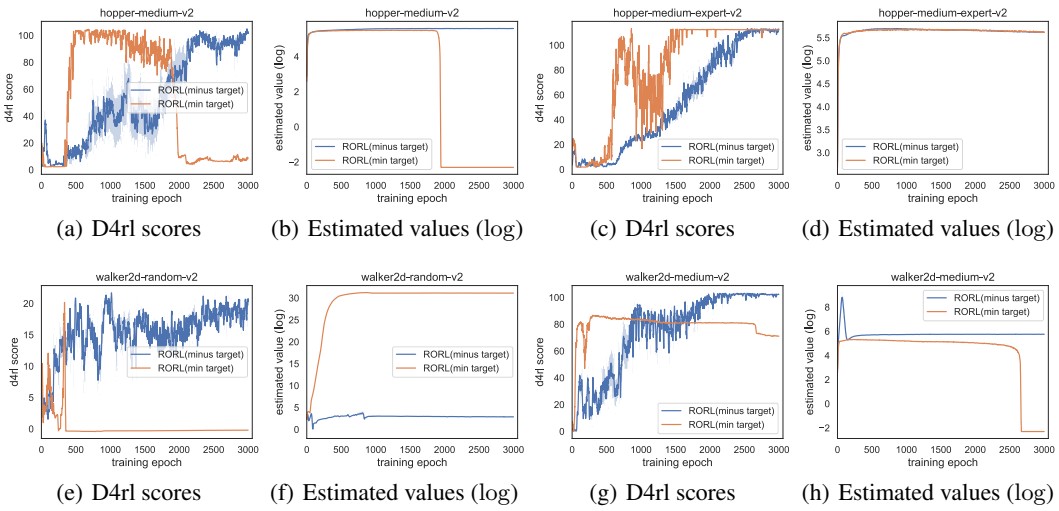

(a) D4rl scores     (b) Estimated values (log)     (c) D4rl scores     (d) Estimated values (log)

(e) D4rl scores     (f) Estimated values (log)     (g) D4rl scores     (h) Estimated values (log)

Figure 19: Comparison of the "minus target" and the "min target" in the OOD loss $\mathcal{L}_{\text{ood}}$ on four tasks.

slightly better on the walker2d-medium task. We think this might be because we train the policy and value networks for $3 \times 10^6$ training steps, which may narrow the gap of the two optimization methods. On the contrary, the mixed-order attackers ('action diff mixed order' and 'min $Q$ mixed order') work better than zeroth-order attackers ('action diff' and 'min $Q$') in the evaluation period, as demonstrated in Figure 4.

### C.14    Comparison of the "minus target" and the "min target"

In the OOD loss $\mathcal{L}_{\text{ood}}$ (Eq. (4)), the pseudo-target $\widehat{\mathcal{T}}_{\text{ood}}Q_{\phi_i}(\hat{s}, \hat{a})$ for the OOD state-action pairs $(\hat{s}, \hat{a})$ can be implemented in two ways to underestimate the values of $(\hat{s}, \hat{a})$: $\widehat{\mathcal{T}}_{\text{ood}}Q_{\phi_i}(\hat{s}, \hat{a}) := Q_{\phi_i}(\hat{s}, \hat{a}) - \lambda u(\hat{s}, \hat{a})$ or $\widehat{\mathcal{T}}_{\text{ood}}Q_{\phi_i}(\hat{s}, \hat{a}) := \min_{i=1,...,K}Q_{\phi_i}(\hat{s}, \hat{a})$ ($K = 10$). The two targets are referred to as the "minus target" and the "min target" respectively. In Figure 19, we compare the two targets' D4RL scores in clean environments, and the hyper-parameters are the same as Table 4. Although the "min target" has less hyper-parameters and achieves comparable performance on the hopper-medium-expert-v2 task, it is unstable and not flexible across different tasks, e.g., significantly overestimating values for the walker2d-random-v2 task and underestimating values for hopper-medium-v2 and walker2d-medium-v2 tasks. Therefore, we choose the "minus target" by default in our paper.

Table 9: Hyper-parameters of RORL for the Adroit domains.

| Task Name | $\beta_{\text{Q}}$ | $\beta_{\text{P}}$ | $\beta_{\text{ood}}$ | $\epsilon_{\text{Q}}$ | $\epsilon_{\text{P}}$ | $\epsilon_{\text{ood}}$ | $\tau$ | $n$ | $\lambda$ (d) |
|---|---|---|---|---|---|---|---|---|---|
| Pen-human
Hammer-human
Door-human
Relocate-human | 0.0001 | 0.01 | 0.5 | 0.001 | 0.001 | 0.001
0.01
0.01
0.01 | 0.2 | 20 | $0.2 \to 0.1\ (1e^{-6})$
$2 \to 0.5\ (2e^{-6})$
$1 \to 0.5\ (1e^{-6})$
$1 \to 0.5\ (1e^{-6})$ |
| Pen-cloned
Hammer-cloned
Door-cloned
Relocate-cloned | 0.0001 | 0.1 | 0.5 | 0.001
0.005
0.001
0.005 | 0.001
0.005
0.001
0.005 | 0.001
0.01
0.01
0.01 | 0.2 | 20 | $1 \to 0.2\ (2e^{-6})$
$2 \to 0.5\ (2e^{-6})$
$1 \to 0.5\ (1e^{-6})$
$2 \to 1.0\ (1e^{-6})$ |
| Pen-expert
Hammer-expert
Door-expert
Relocate-expert | 0.0001 | 1.0 | 0.5 | 0.005
0.005
0.005
0.001 | 0.005
0.005
0.005
0.001 | 0.01 | 0.2 | 20 | $2.0 \to 2.0\ (0.0)$
$1 \to 0.5\ (1e^{-6})$
$1.5 \to 1.5\ (0.0)$
$3 \to 2.0\ (2e^{-6})$ |

## C.15 Experiments in Adroit Domains

We also evaluate RORL in the challenging Adroit domains which control a 24-DoF robotic hand to manipulate a pen, a hammer, a door and a ball. These domains contain three types of data, namely 'Expert', 'Cloned', and 'Human', for each task. The hyper-parameters are listed in Table 9. We set $\beta_Q = 0.0001$, $\beta_{ood} = 0.5$, $\tau = 0.2$, $n = 20$, and search $\beta_P$ within $\{0.01, 0.1, 1.0\}$, $\epsilon_Q/\epsilon_P/\epsilon_{ood}$ within $\{0.001, 0.005, 0.01\}$. For Door/Relocate-human/cloned datasets, the policy learning rate is set to $1e^{-4}$. The other hyper-parameters are the same as in Table 3. On four expert datasets, we train RORL for 1000 epochs (1000 gradient steps per epoch). As for other datasets, we train RORL for 300 epochs because the 'Cloned' and 'Human' datasets are much smaller.

In Table 10, we compare the performance of RORL with other baselines, such as EDAC, PBRL, TD3+BC, CQL, UWAC, BEAR, and BC. We can observe that RORL achieves the top two highest score in 6 out of 12 tasks, which further verifies the effectiveness of RORL.

Table 10: Average normalized score over 3 seeds in Adroit domain. Top two highest scores are highlighted.

|  |  | BC | BEAR | UWAC | CQL | TD3+BC | PBRL | EDAC | RORL |
|---|---|---|---|---|---|---|---|---|---|
| Human | Pen | 34.4 | -1.0 | 10.1 ±3.2 | **37.5** | 0.0 | 35.4 ±3.3 | **52.1±8.6** | 33.7 ± 7.6 |
| | Hammer | 1.5 | 0.3 | 1.2 ±0.7 | **4.4** | 0.0 | 0.4 ± 0.3 | 0.8±0.4 | **2.3 ± 1.9** |
| | Door | 0.5 | -0.3 | 0.4 ±0.2 | **9.9** | 0.0 | 0.1 ±0.0 | **10.7±6.8** | 3.78 ± 0.7 |
| | Relocate | 0.0 | -0.3 | 0.0 ±0.0 | **0.2** | 0.0 | 0.0 ±0.0 | **0.1±0.1** | 0.0 ± 0.0 |
| Cloned | Pen | 56.9 | 26.5 | 23.0 ±6.9 | 39.2 | 0.0 | **74.9 ±9.8** | **68.2±7.3** | 35.7± 3.1 |
| | Hammer | 0.8 | 0.3 | 0.4 ±0.0 | **2.1** | 0.0 | 0.8 ±0.5 | 0.3±0.0 | **1.7 ±0.5** |
| | Door | -0.1 | -0.1 | 0.0 ±0.0 | 0.4 | 0.0 | **4.6 ±4.8** | **9.6±8.3** | -0.1 ± 0.1 |
| | Relocate | -0.1 | -0.3 | -0.3 ±0.0 | -0.1 | 0.0 | -0.1 ±0.0 | **0.0±0.0** | **0.0 ± 0.0** |
| Expert | Pen | 85.1 | 105.9 | 98.2 ±9.1 | 107.0 | 0.3 | **137.7 ±3.4** | 122.8 ± 14.1 | **130.3 ± 4.2** |
| | Hammer | 125.6 | 127.3 | 107.7 ±21.7 | 86.7 | 0.0 | **127.5 ±0.2** | 0.2 ± 0.0 | **132.2 ± 0.7** |
| | Door | 34.9 | 103.4 | **104.7 ±0.4** | 101.5 | 0.0 | 95.7 ±12.2 | -0.3 ± 0.1 | **104.9 ± 0.9** |
| | Relocate | **101.3** | 98.6 | **105.5 ±3.2** | 95.0 | 0.0 | 84.5 ±12.2 | -0.3 ± 0.0 | 47.8 ± 13.5 |

## C.16 AntMaze Tasks

The AntMaze domain is a challenging navigation domain with an 8-DoF Ant quadruped robot and three types of datasets, namely 'umaze', 'medium', and 'large'. In this domain, the agent receives a sparse reward of 0/1, where reward 1 is given only when the ant reaches the desired goal. The challenges for the AntMaze domain are sparse rewards and multitask data, which might be beyond the scope of our study. To the best of our knowledge, very few ensemble-based offline RL algorithms can work in this domain, probably because estimating uncertainty in a sparse reward setting is difficult. A recent work [15] conducted in-depth research on this problem and found that the independent target is crucial for the uncertainty estimation in ensemble-based offline RL. We adopt the techniques used in [15] for RORL and reported the results in Table 12. We only use the OOD loss and policy smoothing loss for RORL, and replace the shared min target in Eq. (1) with the independent target to train $Q$ functions:

$$\widehat{\mathcal{T}}Q_{\phi_i}(s,a) := r(s,a) + \gamma\widehat{\mathbb{E}}_{a'\sim\pi_\theta(\cdot|s')}\big[Q_{\phi_i'}(s',a') - \alpha \cdot \log \pi_\theta(a'|s')\big], \tag{30}$$

In Eq (6), we can train the policy with the 'LCB' objective (i.e., $\text{mean}_{j=1,...,K}Q_{\phi_j}(s,a) - c \cdot \text{std}_{j=1,...,K}Q_{\phi_j}(s,a)$, where $c = 4$ is used in our experiments) or the 'Min' target (i.e., $\min_{j=1,...,K}Q_{\phi_j}(s,a)$) to enforce pessimism. Following [15], we train RORL for $2 \times 10^5$ training steps and evaluate the final performance for 100 episodes. Instead of changing the 0/1 reward to -2/2, we adopt reward shifting [51] to change the 0/1 reward to 0.001/10. We also find that adding the BC loss to the policy loss is helpful for antmaze-umaze tasks. Therefore, we add the BC loss to the policy loss for $5 \times 10^4$ training steps for all tasks, except for antmaze-umaze-diverse, where we add the BC loss for $2 \times 10^5$ training steps. Other hyper-parameters such as the coefficient $\beta_{BC}$ of BC loss are listed in Table 11.

We also apply our policy and value function smoothing techniques on top of IQL (short for 'IQL+smoothing'). For the hyper-parameters, we use $\epsilon_Q = 0.01$, $\epsilon_P = 0.03$, $\tau = 0.2$ for all six types of datasets, and search $\beta_Q \in \{0.1, 0.01\}$, $\beta_P \in \{0.1, 0.5\}$, $n = 20$. Other hyper-parameters keep the default hyper-parameters of IQL [27].

In Table 12, we compare RORL and 'IQL+smoothing' with both model-free (AWAC [37], TD3+BC [13], CQL [29], and IQL [27]) and model-based (ROMI [58]) baselines. RORL achieves the highest average score on the 6 tasks. Besides, on 4 out of 6 tasks, 'IQL+smoothing' improves the performance of IQL. Intuitively, for sparse reward tasks, smoothing the value functions of nearby states could help with the value propagation, and smoothing the policy can enhance the robustness of learned policies. But we can still notice that RORL does not perform well on the antmaze-large task, which may be a future improvement work.

Table 11: Hyper-parameters of RORL for the AntMaze domains.

| Task Name | $\beta_{\mathrm{P}}$ | $\beta_{\mathrm{ood}}$ | $\epsilon_{\mathrm{P}}$ | $\epsilon_{\mathrm{ood}}$ | $n$ | policy objective | $\beta_{\mathrm{BC}}$ | $\lambda\,(d)$ |
|---|---|---|---|---|---|---|---|---|
| umaze | | 0.3 | | | | LCB | | $1.0 \to 1.0\,(0)$ |
| umaze-diverse | | 0.3 | | | | LCB | | $2.0 \to 2.0\,(0)$ |
| medium-play | 1.0 | 0.3 | 0.005 | 0.01 | 20 | LCB | 10 | $1.0 \to 1.0\,(0)$ |
| medium-diverse | | 0.3 | | | | LCB | | $2.0 \to 1.0\,(1e^{-6})$ |
| large-play | | 0.5 | | | | Min | | $2.0 \to 1.0\,(1e^{-6})$ |
| large-diverse | | 0.3 | | | | Min | | $1.0 \to 1.0\,(0)$ |

Table 12: Comparison of final performance on AntMaze tasks. The results are average over 3 random seeds. Top two scores for each task are highlighted.

| | BC | AWAC | TD3+BC | CQL | ROMI+BCQ | IQL | IQL+smoothing | RORL |
|---|---|---|---|---|---|---|---|---|
| antmaze-umaze | 54.6 | 56.7 | 78.6 | 74.0 | $68.7 \pm 2.7$ | 87.5 | **92.3±4.6** | **96.7 ± 1.9** |
| antmaze-umaze-diverse | 45.6 | 49.3 | 71.4 | **84.0** | $61.2 \pm 3.3$ | 62.2 | $64.0 \pm 5.6$ | **90.7±2.9** |
| antmaze-medium-play | 0.0 | 0.0 | 10.6 | 61.2 | $35.3 \pm 1.3$ | 71.2 | **75.3±2.5** | **76.3±2.5** |
| antmaze-medium-diverse | 0.0 | 0.7 | 3.0 | 53.7 | $27.3 \pm 3.9$ | **70.0** | **74.3 ± 3.7** | 69.3±3.3 |
| antmaze-large-play | 0.0 | 0.0 | 0.2 | 15.8 | $20.2 \pm 14.8$ | **39.6** | **38.3 ± 4.8** | 16.3±11.1 |
| antmaze-large-diverse | 0.0 | 1.0 | 0.0 | 14.9 | **41.2 ±4.2** | **47.5** | $40.0 \pm 7.8$ | 41.0±10.7 |
| Average | 16.7 | 17.95 | 27.3 | 50.6 | 42.3 | 63.0 | **64.0** | **65.1** |

### C.17 Robustness of the Benchmark Results

In Figure 20, we evaluate the robustness of the benchmark results, i.e., how robust each algorithm is to maintain the performance listed in Table 1. We compare RORL with EDAC, SAC-10 on six tasks. EDAC is reproduced with 10 ensemble $Q$ networks as RORL and SAC-10, and uses $\eta = 1$ for all six tasks. Note that in the benchmark experiments, RORL is only trained with small smoothing scales within $\{0.001, 0.005, 0.01\}$. The evaluation perturbation scales are within range $[0.00, 0.05]$ and the results are averaged over 4 random seeds. From the results, we can conclude that RORL can successfully keep the highest performance within a certain perturbation scale and the performance of EDAC and SAC-10 decreases faster than RORL for most tasks and attack methods. The results imply that RORL has better practicability in real-world scenarios.

## D Tips for Customizing RORL

According to our ablation study result in Appendix C, we summarize some tips for adapting RORL for customized use below.

- **Hyper-parameter Tuning:** Since RORL is proposed to solve a challenging problem, it has many hyper-parameters. Our first suggestion is to use our hyper-parameter search range in Appendix B.1. You can tune them according to the importance of each component, where the general order is : OOD loss > policy smoothing loss > $Q$ smoothing loss.

- **Computation Cost:** If you want less GPU memory usage and less training time, you can (1) set $\beta_{\mathrm{Q}} = 0$ and $\epsilon_{\mathrm{Q}} = 0$ because the $Q$ smoothing loss contributes the least but consumes a large computational cost, and (2) use a small number $n$ of sampled perturbed states to reduce the GPU memory usage.

# E    More Related Works

**Model-Based Offline RL**    In offline RL, model-based methods use an empirical model learned from the offline dataset to enhance the generalization ability. The model can be used as the virtual environment for data collection [72, 26], or to augment the dataset for an existing model-free algorithm [71, 58]. The main challenges of model-based algorithms are how to learn the accurate empirical model and how to construct the uncertainty measure. A recent work [22] demonstrates that the transformer model can generate realistic trajectories, which is beneficial for policy learning. In contrast, we focus on the model-free methods in this paper and leave the robustness of model-based methods in future work.

**Adversarial Attack**    Inspired by adversarial examples in deep learning [18, 40], adversarial attack and policy poisoning [8, 20, 42] are studied to avoid adversarial manipulations on the network policies. Gleave et al. [16] study adversarial policy in the behavior level [16]. Data corruption [74, 34, 62] considers the case where an attacker can arbitrarily modify the dataset under a specific budget before training. While adversarial attack in RL is highly related to robust RL, they focus more on adversarial attacks compared to our robustness setting. More effective attack strategies for offline RL can facilitate learning more robust policies.

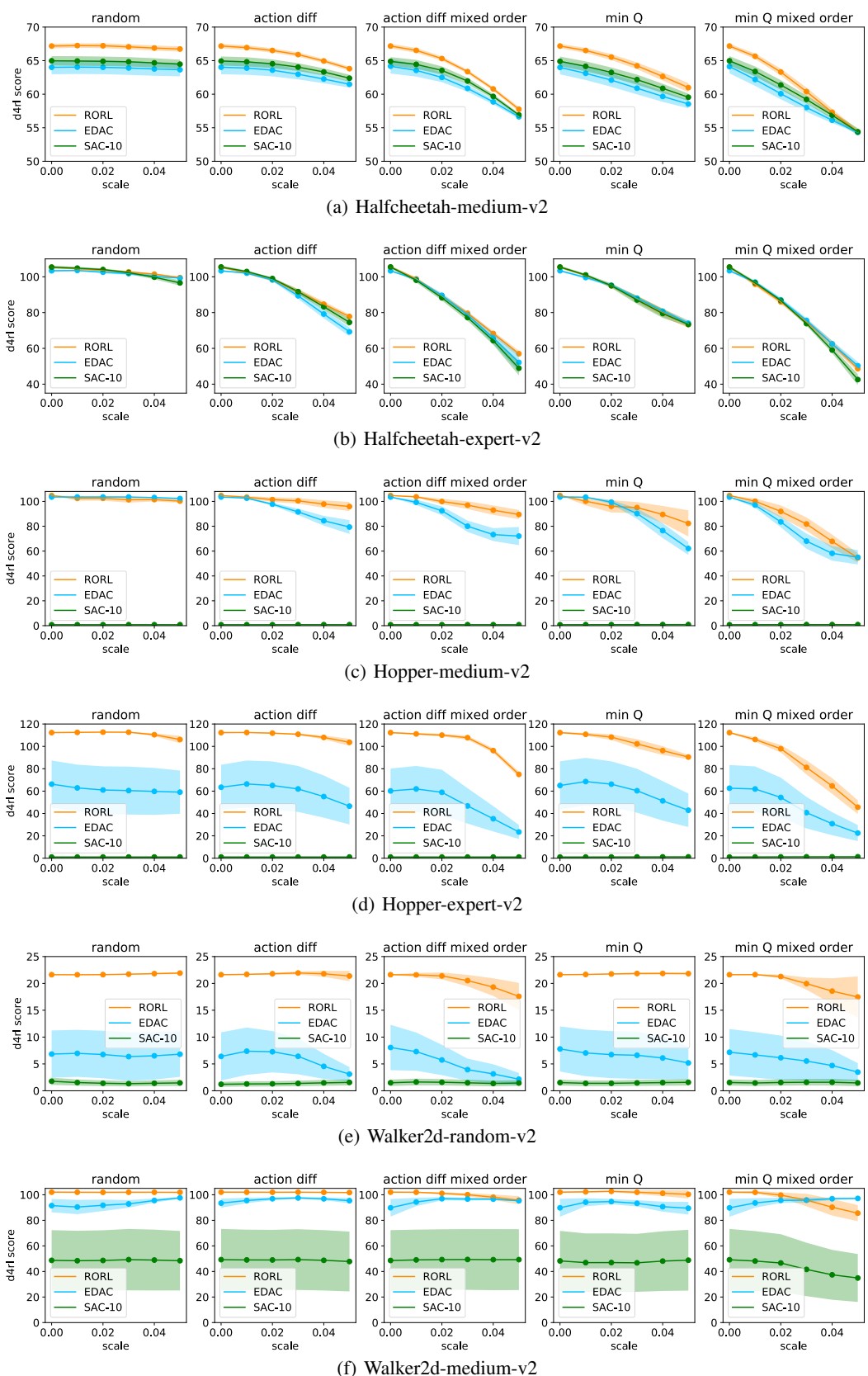

Figure 20: Performance under adversarial attack on six datasets. RORL can maintain the best performance in the benchmark experiments for small-scale perturbations.