# OpenReview forum: "RORL: Robust Offline Reinforcement Learning via Conservative Smoothing"
_NeurIPS.cc/2022/Conference — NeurIPS 2022 Accept_

### Official Review · Reviewer_frMM · 2022-07-10

**Rating:** 5
**Confidence:** 4
**Soundness:** 2 fair
**Presentation:** 3 good
**Contribution:** 2 fair

**Summary:**

The authors handled my major concerns on approximation and experiments by providing additional responses and adding more experiments. I'd like to improve my score as boarderline accept.

===

Training robust RL agent from offline datasets is an important yet challenging problem. This paper proposes RORL: offline RL algorithms with conservative smoothing. The main idea is to add smooth constraints (forcing agent to generate outputs on (adversarially) perturbed inputs) to offline RL algorithm. Here, to avoid overestimation issues, the authors also utilize uncertainties from Q-function as a penalty term. The proposed method not only achieved strong results on mujoco tasks from D4RL datasets but also showed that learned agents are more robust to perturbation.

**Questions:**

Q1. I like the examples for motivating the paper in Section 3 but have the following minor questions:

* To obtain a perturbed state, the authors utilize action diff. However, generating perturbed state with respect to Q-function would be more natural choice for this example. Because the authors considered the perturbation with respect to Q-function in main algorithm and experiments, I wonder if there is a special reason to generate perturbed state using action diff here.

* The author mentioned that they choose one of the reduced dimensions (by PCA) in Figure 3. It would be nice if the authors can clarify how to choose this dimension (for reproducibility).

Q2. I have the following questions about the method:

* What is the definition of perturbation set?

* According to Algorithm 1, the authors mentioned that $\hat s$ is sampled from the perturbation set but description in line 133 sounds like they found the worst case state which maximizes eq (2). This part is bit confusing.

* Finding worst case state, which maximizes eq (2), is computationally expensive. Could the author clarify this part?


**Limitations:**

As pointed out in Section 8, the overhead induced by the proposed method can slow down the training. Even though this can be handled later, it would be nice if the authors also can clarify the training overhead from the proposed method (e.g. comparing training time of RORL and other offline RL algorithms).

**Strengths And Weaknesses:**

# Strength

* Motivation is clear and the proposed method sounds reasonable.

# Weakness

* Lack of evaluation on challenging domains. Even though mujoco tasks are standard benchmark in offline RL, it would be nice if the authors can evaluate on more challenging tasks such as AntMaze or Atari. Also, it would be nice if the author can consider the combination with more state-of-the-art offline RL algorithm [1].

# Overall

* I think this paper studies an important research question and proposes a reasonable solution. Also, the authors showed the gains from the proposed method very clearly on standard offline RL benchmarks. However, at the same time, there are several concerns (i.e., more evaluation on challenging tasks, combination with more stated-of-the-art offline RL algorithms and so on) about the draft. Because of that, I'd like to suggest "weak reject" but I'm also willing to change my score based on other reviews and author responses.

[1] Kostrikov, I., Nair, A. and Levine, S., 2021. Offline reinforcement learning with implicit q-learning. arXiv preprint arXiv:2110.06169.

---

> ### Author Response · Authors · 2022-08-02
> **Response to Reviewer frMM (part 1/2)**
>
> Thanks for your thoughtful comments. We provide clarification to your questions as below. We appreciate it if you have any further questions or comments.
>
> **Q1: It would be nice if the authors can evaluate on more challenging tasks such as AntMaze or Atari. Also, it would be nice if the author can consider the combination with more state-of-the-art offline RL algorithm (IQL).**
>
> A1: Currently, our method is hard to be evaluated in Atari domain since sampling adversarial noises for image-based observation is computationally expensive. Nevertheless, it would be interesting if we sample adversarial noises in the feature space. We leave this for future research.
>
> To solve your concerns, we add experiments on challenging Adroit domain from the D4RL benchmark, and the results are provided in the Appendix C.8. In Table 9, we compare the performance of RORL with baselines such as EDAC, PBRL, TD3+BC, CQL, UWAC, BEAR, and BC. We also list the results below for a quick review. RORL achieves the top two highest score in 6 out of 12 tasks, which further verifies the effectiveness of RORL.
>
> The experiment for combining the smoothing techniques and IQL is provided in Appendix C.7. We find IQL with the smoothing technique slightly improves the robustness on walker2d and hopper tasks, but it has little effect on the halfcheetah task. The results suggest that simply adopting the smoothing technique does not consistently improve the performance in the offline setting. In contrast, RORL introduces additional OOD underestimation based on uncertainty measure, which contributes to learn conservatively smoothed policy and value functions.
>
> | Adroit Tasks | BC | BEAR | UWAC | CQL | TD3+BC | PBRL | EDAC | RORL |
> | :-----| :----: | :----: |:----: |:----: |:----: |:----: |:----: |:----: |
> | Pen-Human  | 34.4 | -1.0 |   10.1  $\pm$3.2 |   **37.5** |  0.0 |   35.4  $\pm$3.3 |  **52.1$\pm$8.6** | 33.7 $\pm$ 7.6 |
> | Hammer-Human|  1.5 | 0.3 |    1.2 $\pm$0.7 |   **4.4** |  0.0 |   0.4  $\pm$ 0.3 | 0.8$\pm$0.4 |  **2.3 $\pm$ 1.9** |
> | Door-Human | 0.5 | -0.3 |   0.4  $\pm$0.2 |   **9.9** |  0.0 |   0.1  $\pm$0.0  | **10.7$\pm$6.8**  | 3.78 $\pm$ 0.7 |
> | Relocate-Human | 0.0 | -0.3 |   0.0  $\pm$0.0 |   **0.2** |  0.0 |   0.0  $\pm$0.0 | **0.1$\pm$0.1**   |  0.0 $\pm$ 0.0 |
> | Pen-Cloned  |   56.9 | 26.5 |   23.0  $\pm$6.9 | 39.2 |  0.0 |   **74.9 $\pm$9.8** | **68.2$\pm$7.3** | 35.7$\pm$ 3.1 |
> | Hammer-Cloned | 0.8 | 0.3 |   0.4 $\pm$0.0 |   **2.1**|  0.0 |   0.8  $\pm$0.5  |  0.3$\pm$0.0    | **1.7 $\pm$0.5** |
> | Door-Cloned | -0.1 | -0.1 |   0.0 $\pm$0.0 | 0.4 |  0.0 |  **4.6   $\pm$4.8** |  **9.6$\pm$8.3**   | -0.1 $\pm$ 0.1|
> | Relocate-Cloned | -0.1 | -0.3 |   -0.3  $\pm$0.0 | -0.1 |  0.0 |   -0.1   $\pm$0.0   |  **0.0$\pm$0.0**    | **0.0 $\pm$ 0.0** |
> | Pen-Expert  | 85.1 | 105.9 |   98.2   $\pm$9.1 | 107.0 |  0.3 |   **137.7   $\pm$3.4** | 122.8 $\pm$ 14.1  | **130.3 $\pm$ 4.2**|
> | Hammer-Expert  |   125.6 |   127.3 |   107.7   $\pm$21.7 | 86.7 |  0.0 |  **127.5  $\pm$0.2** | 0.2 $\pm$ 0.0 | **132.2 $\pm$ 0.7** |
> | Door-Expert | 34.9 | 103.4 |   **104.7  $\pm$0.4** | 101.5 |  0.0 |   95.7  $\pm$12.2 | -0.3 $\pm$ 0.1   | **104.9 $\pm$ 0.9** |
> | Relocate-Expert  |   **101.3** |   98.6 |   **105.5  $\pm$3.2** | 95.0 |  0.0 |   84.5  $\pm$12.2 | -0.3 $\pm$ 0.0 | 47.8 $\pm$ 13.5 |
>
> **Q2: To obtain a perturbed state, the authors utilize action diff. However, generating perturbed state with respect to Q-function would be more natural choice for this example. Because the authors considered the perturbation with respect to Q-function in main algorithm and experiments, I wonder if there is a special reason to generate perturbed state using action diff here.**
>
> A2: We use 'action diff' since it is faster to optimize than the 'min Q' attack. We provide additional results with 'min Q mixed order' attack in Figure 6 of Appendix B.3. The result shows that CQL-smooth also shows better robustness than CQL on the performance and the learned Q functions.
>
> **Q3: The author mentioned that they choose one of the reduced dimensions (by PCA) in Figure 3. It would be nice if the authors can clarify how to choose this dimension (for reproducibility).**
>
> A3: For visualizing the relationship between the $Q$-function and the state space, we sample 2560 adversarial transitions for each attack $\epsilon$ and calculate the corresponding $Q$-function. Since the state has relatively high dimensions (i.e., 11 or 17), we perform PCA dimensional reduction to reduce the state to 4 dimensions. We find the $Q$-function generally has a strong correlation to one or two dimensions of the state after dimensional reduction. For other dimensions, the relationship between the $Q$-value and the PCA-reduced state often has one or two peaks, which has less variety in the curve. **We add more discussions for  reproducibility in Appendix B.3**.

---

> > ### Author Response · Authors · 2022-08-02
> > **Response to Reviewer frMM (part 2/2)**
> >
> > **Q4: What is the definition of perturbation set?**
> >
> > A4: The perturbation set $\mathbb{B}_{d}(s,\epsilon)=\{\hat{s}:d(s,\hat{s})\leq \epsilon\}$ for state $s$ is defined as an $\epsilon$-radius ball measured in metric $d(·,·)$, which is chosen to be the $L_p$ distance, and we use $L_\infty$ in our paper. We also updated the manuscript to more clearly emphasize the definition in Section 2, Section 3 and Section 4.
> >
> >
> > **Q5: According to Algorithm 1, the authors mentioned that s is sampled from the perturbation set but description in line 133 sounds like they found the worst case state which maximizes eq (2). This part is bit confusing.**
> >
> > A5: Directly maximizing Eq.(2) to find the best attack is computationally expensive since the perturbation set is too large. Therefore, we uniformly sample 50 states in the perturbation set $\mathbb{B}_{d}(s,\epsilon)$, and then select the best attack scored by Eq. (2) from these sampled states, which has the advantage of simplicity and less computation cost. We make it more clear in the revision.
> >
> > **Q6: Finding worst-case state, which maximizes eq (2), is computationally expensive. Could the author clarify this part?**
> >
> > A6: Maximizing Eq.(2) to find the optimal attack needs to take gradient descent for many steps starting from a randomly initialized point. Such process needs to be performed for each sampled state, which is computationally expensive. In RORL, we use zero-order optimization by first sampling several examples within $B_d(s, \epsilon)$, and then choosing the one maximizes Eq. (2) as the adversarial perturbation.
> >
> > We remark that such simplification is only used in training. We perform the standard optimization process for adversarial attack including 'min Q mixed order' and 'action diff mixed order' in the evaluation period.
> >
> > **Q7: As pointed out in Section 8, the overhead induced by the proposed method can slow down the training. Even though this can be handled later, it would be nice if the authors also can clarify the training overhead from the proposed method (e.g. comparing training time of RORL and other offline RL algorithms).**
> >
> > A7: We compare the training time between RORL and other methods in Appendix C.5. For each method, we measure the average epoch time and the GPU memory usage on the hopper-medium-v2 task.
> >
> > As shown in Table 8, RORL runs slightly faster than CQL, mainly because CQL needs the OOD action sampling and the logsumexp approximation. For ensemble-based baselines, RORL runs much faster than PBRL, requiring only 28.7$\%$ of PBRL's epoch time. In RORL, we also include the OOD state-action sampling and additional adversarial training procedures, but we implemented these procedures efficiently based on GPU operation and parallelism. Even so, RORL is still slower than SAC-10 and EDAC. As for the GPU memory consumption, RORL uses comparable memory to PBRL and EDAC, with only $16.7\\%$ more memory usage. We also list the average epoch time below for a quick review.
> >
> > |  | CQL |SAC-10| EDAC |PBRL|RORL|
> > | :-----|  :----: | :----: | :----: | :----: |:----: |
> > | Average Epoch Time (s)| 32.40 | 12.73 | 17.94 | 102.96| 29.56|
> > | GPU Memory (GB)  | 1.4 | 1.3|   1.8  | 1.8| 2.1|
> >
> > In the future, we would like to research on accelerating the training process of RORL.

---

> > > ### Comment · Reviewer_frMM · 2022-08-07
> > > **response**
> > >
> > > Thank you for the detailed answers.
> > >
> > > After reading rebuttals and other reviews, I still think that this work is on borderline due to the following reasons:
> > >
> > > (1) Method: the authors clarified the training process as follows: `we uniformly sample 50 states in the perturbation set , and then select the best attack scored by Eq. (2) from these sampled states`. To be honest, I don't think this is a right way to generate adversarial samples, which means that current implementation is not exactly optimizing the proposed objective. Regardless of experimental results, it would be nice if the authors verified the proposed ideas in more principled way...
> > >
> > > (2) No significant gains on Adroit: I really appreciate the author's efforts for additional experiments but the poor performances on new tasks raise some concerns about the proposed ideas. Also, I think the evaluation on AntMaze is more convincing b/c it is the domain to see the real gains from offline RL due to reward sparsity.

---

> > > > ### Author Response · Authors · 2022-08-08
> > > > **Response to Reviewer frMM**
> > > >
> > > > Thank you for your response. We still would like to try to clarify a few points in your concerns:
> > > >
> > > > **Q1: Optimization of the objectives**
> > > >
> > > > **A1**: Solving the inner maximization of a min-max problem perfectly to generate adversarial perturbations is usually impractical for non-convex problems. Prior studies leverage both zeroth-order [1][2][3] and first-order optimization [4][5] methods to approximately solve this problem. **Zeroth-order optimization is generally useful, especially for black-box settings where we can only access the input and output of neural networks without explicit gradient information**. Therefore, we chose a kind of zeroth-order optimization to solve the objective. In our experiments (in Figure 4), we demonstrate that both zeroth-order attackers (“action diff” and “min Q”) are more effective than the random attacker. Moreover, we think our work might also inspire future directions on black-box adversarial attacks for RL.
> > > >
> > > > **We use a sampling-based zeroth-order optimization in the training phase to reduce the computation cost for RORL**. In fact, we also implemented a first-order version of RORL, which requires an average epoch time of **72.7s** on V100 GPU (while the average epoch time of the zeroth-order method is **29.6s**). We didn’t observe any significant gain on robustness in the first-order version (more empirical results of the first-order version will be added soon).  In our experiments (Figure 4), we also show that RORL with the current optimization procedure can already defend against the strong attack of first-order attackers (“action diff mixed order” and “min Q mixed order”). So we think using the sampling-based zeroth-order optimization to approximate the solution of the original problem is reasonable.
> > > >
> > > > In summary, we believe **our method is a valid approximation of generating adversarial perturbations during training, which has the advantage of little computational cost and enjoys comparable robustness with the first-order version**. Since there is little prior work on adversarial attack in the offline RL setting, we think there are still rooms to improve on the optimization method and the adversarial perturbation generation.
> > > >
> > > >
> > > > **Q2: No significant gains on Adroit ... the Antmaze tasks are more convincing.**
> > > >
> > > > **A2:** We agree that the AntMaze benchmark post challenges with sparse-reward setting for current offline RL algorithms. To be frank, we were also curious if RORL could work well on such settings, while we think that **handling sparse rewards and improving policy robustness are orthogonal directions in investigating new offline RL algorithms**. In this paper, we focus on the policy robustness under adversarial attack, and we think considering the most general tasks commonly used for adversarial attack by prior online robust RL works [4][5][6] should be sufficient. In our experiments, we have already shown that RORL significantly improves the robustness under adversarial observation perturbation over current SOTA offline RL algorithms. We think our experimental results over 15 Mujoco tasks and 12 Adroit tasks are sufficient to demonstrate that RORL can achieve SOTA or comparable performance on clean environments.
> > > >
> > > > Thank you again for your effort in making our paper better!
> > > >
> > > >
> > > > [1] Andriushchenko M, Croce F, Flammarion N, et al. Square attack: a query-efficient black-box adversarial attack via random search. ECCV, 2020.
> > > >
> > > > [2] Tu C C, Ting P, Chen P Y, et al. Autozoom: Autoencoder-based zeroth order optimization method for attacking black-box neural networks. AAAI 2019.
> > > >
> > > > [3] Zhang Y, Yao Y, Jia J, et al. How to Robustify Black-Box ML Models? A Zeroth-Order Optimization Perspective. ICLR 2022.
> > > >
> > > > [4] Shen Q, Li Y, Jiang H, et al. Deep reinforcement learning with robust and smooth policy. ICML, 2020.
> > > >
> > > > [5] Zhang H, Chen H, Xiao C, et al. Robust deep reinforcement learning against adversarial perturbations on state observations. NeurIPS, 2020.
> > > >
> > > > [6] Oikarinen T, Zhang W, Megretski A, et al. Robust deep reinforcement learning through adversarial loss. NeurIPS 2021.

---

> > > > > ### Author Response · Authors · 2022-08-09
> > > > > **Response to Reviewer frMM**
> > > > >
> > > > > To verify whether smoothing benefits policy learning in the challenging AntMaze setting, we apply our policy and value function smoothing techniques on top of IQL (short for 'IQL+smoothing').
> > > > >
> > > > > In the following table, we compare 'IQL+smoothing' with both model-free (AWAC [1], TD3+BC [2], CQL [3], and IQL [5]) and model-based (ROMI [4]) baselines. **On 4 out of 6 tasks, 'IQL+smoothing' improves the performance of IQL**. Intuitively, for sparse reward tasks, smoothing the value functions of nearby states could help with value propagation, and smoothing the policy can enhance the robustness of learned policies. We remark that additional penalties for OOD states may also be in need, as discussed in our paper. We would like to provide the results of RORL on the AntMaze task soon.
> > > > >
> > > > >
> > > > > |      | BC|  AWAC | TD3+BC| CQL | ROMI+BCQ|  IQL |   IQL+smoothing |
> > > > > |:-----|:---------:|:-------------:|:------------------:|:----------------:|:--------------:|:----------------:|:----------------:|
> > > > > | antmaze-umaze |             54.6 |               56.7 |      78.6 |      74.0   |   68.7$\pm$2.7 |            87.5 |   **92.3** $\pm$4.6   |
> > > > > |antmaze-umaze-diverse  |     45.6 |              49.3 |      71.4 |      **84.0** |    61.2 $\pm$ 3.3  |            62.2 |   64 $\pm$ 5.6  |
> > > > > | antmaze-medium-play  |     0.0 |             0.0 |        10.6|    61.2 |       35.3 $\pm$1.3 |            71.2 |  **75.3$\pm$2.5** |
> > > > > | antmaze-medium-diverse  |    0.0 |            0.7 |        3.0|    53.7 |       27.3 $\pm$3.9   |            70.0 |   **74.3 $\pm$ 3.7**    |
> > > > > | antmaze-large-play  |      0.0 |            0.0 |          0.2|  15.8 |       20.2 $\pm$ 14.8   |            **39.6** |    38.3 $\pm$ 4.8   |
> > > > > | antmaze-large-diverse  |     0.0 |            1.0 |          0.0|  14.9 |       41.2 $\pm$4.2   |            **47.5** |    40.0 $\pm$ 7.8  |
> > > > >
> > > > > [1] Nair A, Gupta A, Dalal M, et al. Awac: Accelerating online reinforcement learning with offline datasets. arXiv preprint, 2020.
> > > > >
> > > > > [2] Fujimoto S, Gu S S. A minimalist approach to offline reinforcement learning. NeurIPS, 2021.
> > > > >
> > > > > [3] Kumar A, Zhou A, Tucker G, et al. Conservative q-learning for offline reinforcement learning. NeurIPS, 2020.
> > > > >
> > > > > [4] Wang J, Li W, Jiang H, et al. Offline reinforcement learning with reverse model-based imagination. NeurIPS, 2021.
> > > > >
> > > > > [5] Kostrikov I, Nair A, Levine S. Offline Reinforcement Learning with Implicit Q-Learning. ICLR 2022.

---

> > > > > ### Author Response · Authors · 2022-08-09
> > > > > **Comparison of Zeroth-order and First-order Perturbation Generation**
> > > > >
> > > > > We now includes additional comparison between the zeroth-order and first-order perturbation generation in Appendix C.11 and Figure 17.
> > > > > **We hope our response could address your concerns. Thank you again for your time and efforts!**
> > > > >
> > > > > From Figure 17, we can conclude that the two types of optimization for perturbation generation have very similar performance. On halfcheetah-medium task, the first-order version performs slightly better than the zeroth-order version, while the zeroth-order version works slightly better on the walker2d-medium task. **We also list the 'weighted robust score' (i.e., the weighted areas under the performance curves in Figure 16, detailed definition is in Appendix C.4) below for a quick review**. We believe our method is a valid approximation of generating adversarial perturbations during training, which has the advantage of little computational cost and enjoys comparable robustness with the first-order version.
> > > > >
> > > > > |        halfcheetah-medium            |   random |   action diff |   action diff mixed order |   min Q |   min Q mixed order |   Average Score |
> > > > > |:-------------------|:-----------------:|:----------------------:|:----------------------------------:|:----------------:|:----------------------------:|:----------------:|
> > > > > | RORL               |             **57.4** |                  44.5 |                              29.7 |            37   |                        17.7 |            37.2 |
> > > > > | RORL (first-order) |             56.9 |                  **45.3** |                              **32.5** |            **37.4** |                        **19.2** |            **38.2** |
> > > > >
> > > > > |         walker2d-medium           |   random |   action diff |   action diff mixed order |   min Q |   min Q mixed order |   Average Score |
> > > > > |:-------------------|:-----------------:|:----------------------:|:----------------------------------:|:----------------:|:----------------------------:|:----------------:|
> > > > > | RORL               |             94.1 |                  89.1 |                              39.1 |            **61.8** |                        **26.7** |           **62.2** |
> > > > > | RORL (first-order) |             **94.8** |                  **90.1** |                              **42.1** |            53.3 |                        23.9 |            60.8 |

---

> ### Author Response · Authors · 2022-08-06
> **Looking forward to further feedback**
>
> Dear Reviewer frMM,
>
> Since the author-reviewer discussion period has started for a few days, we will appreciate if you could check our response to your review comments soon. This way, if you have further questions and comments, we can still reply before the author-reviewer discussion period ends. If our response resolves your concerns, we kindly ask you to consider raising the rating of our work. Thank you very much for your time and efforts!
>
> Best,
>
> The authors

---

### Official Review · Reviewer_KVko · 2022-07-10

**Rating:** 6
**Confidence:** 3
**Soundness:** 2 fair
**Presentation:** 3 good
**Contribution:** 3 good

**Summary:**

The authors propose an approach to offline reinforcement learning that is robust to small perturbations in the observation space such that the changes are not detrimental to the performance of the final policy. The achieve this by encouraging the value estimator network to be smooth over the state space while being conservative on out of distribution samples. Moreover the learned policy is also constrained to change less with these perturbations.  Experimental results show that the proposed algorithm is able to perform competitively with current Robust / Baseline approaches, and enjoy increased robustness over adversarial attacks.

**Questions:**

Q1: I could not find a definition for the distance metric d used to calculate the epsilon (line 90). Is this a L2 distance metric ?
Q2: Can the epsilon be normalized so that it is takes into account the overall variation of the state space as well as the dimensionality. ?
Q3: What parts are highlighted in the table 1, is this discounting the scores that are not statistically significant during ranking?
Q4: Should the score for scale 0 attack at figure 4(b) be same for all attacks and match the evaluation score at table 1? If not what is the effect of scale zero on the final policy.

**Ethics Review Area:**

["I don’t know"]

**Limitations:**

While the authors clearly mention the adversarial state sampling as their main limitation, they do not quantify this slow down for different state sampling approaches. It would be interesting to see the actual effect on the compute time as a percentage of the total training time.


**Strengths And Weaknesses:**

### Strengths
S1: Overall the motivations for the approach are intuitive and easy to follow. The paper is well written and clear. (excluding some minor comments in the questions section.)

S2: The proposed attack metrics are diverse and well defined under the current framework. The experimental results show that RORL is more robust towards the proposed attacks as compared to other methods while being able to perform competitively under normal conditions.

S3: Theoretically, the proposed framework (RORL) enjoys a tighter suboptimality bound that PBRL.

### Weakness

C1: My main complain towards the paper is that while the authors specifically tackle the tradeoff of conservatism and robustness in offline RL no clear metric has been defined to quantify the robustness of an approach. Hence the reader is forced to trust their eyes over the evaluation curves to judge the robustness of the approach. It may be insightful to invest some thoughts on quantifying the robustness by for example measuring the area under the curve on the performance under attack curve where the scale is normalized over the overall variation and dimensionality of the given dataset.

C2: While this might come off as a knee jerk comment, It would be interesting to see the same set of experiments on more challenging benchmark such as ant-maze, especially as the proposed approach is claiming to better generalization ability and improve the overall robustness of the policy. Will the generalization ability result in significant improvements in a more challenging domain ?

---

> ### Author Response · Authors · 2022-08-02
> **Response to Reviewer KVko (part 1/2)**
>
> Thanks for your constructive comments. We provide clarification to your questions and concerns as below. We appreciate it if you have any further questions or comments.
>
> **Q1: My main complain towards the paper is that no clear metric has been defined to quantify the robustness of an approach, for example measuring the area under the curve on the performance under attack curve where the scale is normalized of the given dataset.**
>
> A1: In prior works[2][3] , the authors only demonstrate the robustness of algorithms via comparing the return curves with different attack scale. To better measure the robustness of RL algorithms, we consider robust score as the areas under the perturbation curve in Appendix C.4. Since the returns have been normalized, we can simply define the **robust score** for each attack strategy as:
> robust score$ = \frac{1}{N} \sum_{i\in [1,N]} Rs[i]$, where $Rs$ is the list of returns under $N$ different attack scales.
>
> The introduced robust score treats different attack scales equally. However, in many real scenarios, we would pay more attention to larger-scale disturbances. To this end, we also define a **weighted robust score** as: $\frac{2}{(1+N)\times N} \sum_{i\in [1,N]} i \times Rs[I]$,
> where the weight is assigned according to the scale order. In Table 6 and Table 7 (Appendix C.4), RORL consistently outperms EDAC and SAC-$10$ on the two robustness metrics. For walker2d and hopper tasks, RORL surpasses EDAC by more than 10 points on both the robust score and the weighted robust score. We also list the average 'weighted robust scores' over five types attack below. More details can be found in Appendix C.4.
>
> |  | halfcheetah-medium| walker2d-medium | hopper-medium |
> | :-----| :----: | :----: |:----: |
> | SAC-10 | 33.4 | 24.7 |1.03 |
> | EDAC | 32.2 | 46.2 |37.0 |
> | RORL | 37.2 | 62.2 |47.7 |
>
>
> [2] Shen Q, Li Y, Jiang H, et al. Deep reinforcement learning with robust and smooth policy. ICML, 2020.
>
> [3] Zhang H, Chen H, Xiao C, et al. Robust deep reinforcement learning against adversarial perturbations on state observations. NeurIPS, 2020.
>
> **Q2: More challenging benchmark such as ant-maze. Will the generalization ability result in significant improvements in a more challenging domain ?**
>
> A2: In Appendix C.8, we further include 12 challenging Adroit tasks from D4RL [5]. In Table 9, we compare the performance of RORL
> with other baselines, such as EDAC, PBRL, TD3+BC, CQL, UWAC, BEAR, and BC. We also list the results below for a quick review. We can observe that RORL achieves the top two highest score in 6 out of 12 tasks, which further verifies the effectiveness of RORL.
>
> | Adroit Tasks | BC | BEAR | UWAC | CQL | TD3+BC | PBRL | EDAC | RORL |
> | :-----| :----: | :----: |:----: |:----: |:----: |:----: |:----: |:----: |
> | Pen-Human  | 34.4 | -1.0 |   10.1  $\pm$3.2 |   **37.5** |  0.0 |   35.4  $\pm$3.3 |  **52.1$\pm$8.6** | 33.7 $\pm$ 7.6 |
> | Hammer-Human|  1.5 | 0.3 |    1.2 $\pm$0.7 |   **4.4** |  0.0 |   0.4  $\pm$ 0.3 | 0.8$\pm$0.4 |  **2.3 $\pm$ 1.9** |
> | Door-Human | 0.5 | -0.3 |   0.4  $\pm$0.2 |   **9.9** |  0.0 |   0.1  $\pm$0.0  | **10.7$\pm$6.8**  | 3.78 $\pm$ 0.7 |
> | Relocate-Human | 0.0 | -0.3 |   0.0  $\pm$0.0 |   **0.2** |  0.0 |   0.0  $\pm$0.0 | **0.1$\pm$0.1**   |  0.0 $\pm$ 0.0 |
> | Pen-Cloned  |   56.9 | 26.5 |   23.0  $\pm$6.9 | 39.2 |  0.0 |   **74.9 $\pm$9.8** | **68.2$\pm$7.3** | 35.7$\pm$ 3.1 |
> | Hammer-Cloned | 0.8 | 0.3 |   0.4 $\pm$0.0 |   **2.1**|  0.0 |   0.8  $\pm$0.5  |  0.3$\pm$0.0    | **1.7 $\pm$0.5** |
> | Door-Cloned | -0.1 | -0.1 |   0.0 $\pm$0.0 | 0.4 |  0.0 |  **4.6   $\pm$4.8** |  **9.6$\pm$8.3**   | -0.1 $\pm$ 0.1|
> | Relocate-Cloned | -0.1 | -0.3 |   -0.3  $\pm$0.0 | -0.1 |  0.0 |   -0.1   $\pm$0.0   |  **0.0$\pm$0.0**    | **0.0 $\pm$ 0.0** |
> | Pen-Expert  | 85.1 | 105.9 |   98.2   $\pm$9.1 | 107.0 |  0.3 |   **137.7   $\pm$3.4** | 122.8 $\pm$ 14.1  | **130.3 $\pm$ 4.2**|
> | Hammer-Expert  |   125.6 |   127.3 |   107.7   $\pm$21.7 | 86.7 |  0.0 |  **127.5  $\pm$0.2** | 0.2 $\pm$ 0.0 | **132.2 $\pm$ 0.7** |
> | Door-Expert | 34.9 | 103.4 |   **104.7  $\pm$0.4** | 101.5 |  0.0 |   95.7  $\pm$12.2 | -0.3 $\pm$ 0.1   | **104.9 $\pm$ 0.9** |
> | Relocate-Expert  |   **101.3** |   98.6 |   **105.5  $\pm$3.2** | 95.0 |  0.0 |   84.5  $\pm$12.2 | -0.3 $\pm$ 0.0 | 47.8 $\pm$ 13.5 |
>
> [5]Fu J, Kumar A, Nachum O, et al. D4rl: Datasets for deep data-driven reinforcement learning. 2020.
>
> **Q3: I could not find a definition for the distance metric d used to calculate the epsilon (line 90). Is this a L2 distance metric ?**
>
> A3: The distance metric $d$ was introduced in Section 2. The perturbation set $\mathbb{B}_{d}(s,\epsilon)=\{\hat{s}:d(s,\hat{s})\leq \epsilon\}$ for state $s$ is defined as an $\epsilon$-radius ball measured in metric $d(·,·)$, which is chosen to be the $L_p$ distance, and we use $L_\infty$ in our paper. We revised the manuscript to more clearly emphasize the definition in Section 2, Section 3 and Section 4.

---

> > ### Author Response · Authors · 2022-08-02
> > **Response to Reviewer KVko (part 2/2)**
> >
> > **Q4: Can the epsilon be normalized so that it is takes into account the overall variation of the state space as well as the dimensionality. ?**
> >
> > A4: In Section 6 (line 221), we mentioned that "To assign uniform adversarial attack budget on each dimension of observations, we normalize the observations for SAC-10, EDAC and RORL.". We record the mean and variance of state in offline data for all tasks, and normalize the sampled states in training. The adversarial noise is added to the normalized state. To emphasize this, we updated the manuscript to more clearly emphasize it in line 94 of Section 3.
> >
> > **Q5: What parts are highlighted in the table 1, is this discounting the scores that are not statistically significant during ranking?**
> >
> > A5: In Table 1, we now highlight the top two highest average normalized scores for each task to show the top-level algorithms for each dataset.
> >
> > **Q6: Should the score for scale 0 attack at figure 4(b) be same for all attacks and match the evaluation score at table 1? If not what is the effect of scale zero on the final policy.**
> >
> > A6: Thanks for your suggestion. The scores for scale 0 attack are the same, but we smoothed the curves with a window size of 3 to better visualize the results. Therefore, the left-most point of each curve in the figure is smoothed and biased from its real value. To alleviate this, we now keep the first and the last points unchanged, and only smooth other points. We will update the figures in the revision.
> >
> > **Q7:  They do not quantify this slow down for different state sampling approaches. It would be interesting to see the actual effect on the compute time as a percentage of the total training time.**
> >
> > A7: We compare the training time between RORL and other methods in Appendix C.5. For each method, we measure the average epoch time and the GPU memory usage on the hopper-medium-v2 task. We also analyze the computational cost of RORL’s three components (Q smooth, policy smooth, and the OOD loss) in Figure 12 of Appendix C.5.
> >
> > As shown in Table 8, RORL runs slightly faster than CQL, mainly because CQL needs the OOD action sampling and the logsumexp approximation. For ensemble-based baselines, RORL runs much faster than PBRL, requiring only 28.7$\\%$ of PBRL's epoch time. In RORL, we also include the OOD state-action sampling and additional adversarial training procedures, but we implemented these procedures efficiently based on GPU operation and parallelism. Even so, RORL is still slower than SAC-10 and EDAC. As for the GPU memory consumption, RORL uses comparable memory to PBRL and EDAC, with only $16.7\\%$ more memory usage.

---

> ### Author Response · Authors · 2022-08-06
> **Looking forward to further feedback**
>
> Dear Reviewer KVko,
>
> Since the author-reviewer discussion period has started for a few days, we will appreciate if you could check our response to your review comments soon. This way, if you have further questions and comments, we can still reply before the author-reviewer discussion period ends. If our response resolves your concerns, we kindly ask you to consider raising the rating of our work. Thank you very much for your time and efforts!
>
> Best,
>
> The authors

---

> > ### Author Response · Authors · 2022-08-07
> > **Dear Reviewer**
> >
> > Since there are only 2 days left in the author-reviewer discussion period, we would appreciate if you could check our reply to your comments soon. We believe that we have addressed the concerns mentioned in your review, please let us know if you have additional concerns.
> >
> > Thank you again for your time and efforts!

---

> > > ### Author Response · Authors · 2022-08-08
> > > **Dear Reviewer**
> > >
> > > We thank you again for your time and efforts! As the response system will be closed soon within 1 days, we wonder whether our responses resolve your concerns? If there are no more questions, we would appreciate if you could kindly consider raising the score. Thank you!

---

> > > > ### Author Response · Authors · 2022-08-09
> > > > **Dear Reviewer KVko**
> > > >
> > > > Since the end of the reviewer-author discussion is reaching, we would appreciate it if you could check our reply to your comments. Thank you very much for your time and efforts!

---

### Official Review · Reviewer_yLKL · 2022-07-11

**Rating:** 6
**Confidence:** 4
**Soundness:** 3 good
**Presentation:** 3 good
**Contribution:** 2 fair

**Summary:**

The paper investigates the problem of training robust RL agents with offline datasets. The paper claims that regularizing policy and value networks to have similar values against adversarial perturbations and applying this technique to PBRL can achieve state-of-the-art performance on the D4RL benchmarks in both standard and adversarial settings. The paper also provides a theoretical analysis of the sub-optimality gap of the proposed algorithm in linear MDPs.

**Questions:**

**OOD penalization**

As mentioned in line 147, the difference between the proposed method and PBRL in OOD penalization is that the proposed method penalizes both OOD states and actions. However, assuming that $\hat{a}$ is close to $a$, the proposed method has a similar effect as PBRL since the Q-functions are trained to have similar values for $s$ and $\hat{s}$. Could you please conduct an ablation study that only penalizes OOD actions like PBRL?

**Expert datasets**

Is there any reason to exclude *-expert-v2 or *-medium-expert-v2 datasets from the experiments?

**Comparison with S4RL**

I think the paper is closely related to S4RL [1], which utilizes adversarial examples for smoothness. It would be better to provide the results of CQL + S4RL or PBRL + S4RL.

[1] Sinha, et al., S4RL: Surprisingly Simple Self-Supervision for Offline Reinforcement Learning in Robotics, 2021.

**Limitations:**

I think the proposed method is much slower than other offline RL algorithms since it has to solve mini-max problems. Please report the wall-clock time of the proposed method in the main text.

**Strengths And Weaknesses:**

Strengths:
- The paper thoroughly analyses the proposed algorithm both empirically and theoretically.
- The proposed method achieves state-of-the-art performance in the standard offline RL benchmark, while the performance improvement is quite marginal.

Weaknesses:
- The paper does not provide experimental results on expert or near-expert datasets.
- The proposed algorithm requires computing an adversarial example $\hat{s}$, which is computationally expensive.

---

> ### Author Response · Authors · 2022-08-02
> **Response to Reviewer yLKL (part 1/2)**
>
> Thanks for your insightful feedback. We provide clarification to your questions as below. We appreciate it if you have any further questions or comments.
>
> **Q1: The paper does not provide experimental results on expert or near-expert datasets. Is there any reason to exclude expert-v2 or medium-expert-v2 datasets from the experiments?**
>
> A1: In our original submission, we already conduct experiments on the three **medium-expert** datasets in Section 6.1 and Table 1. We also include three **expert** datasets in Table 1 in the revised version. We list the results on "medium-expert" (short for '-m-e') and 'expert' (short for '-e') datasets below for a quick review. RORL consistently achieves the SOTA performance on both types of datasets.
>
> |      |   BC |   CQL |   PBRL|   SAC-10   |   EDAC(paper) |   EDAC-10 | RORL |
> |:-----|-----------------:|----------------------:|----------------------------------:|----------------:|----------------------------:|----------------:|----------------:|
> |halfcheetah-m-e | 44.0$\pm$1.6  |  95.0$\pm$1.4 | 92.3$\pm$1.1 | 107.1$\pm$2.0 | 106.3$\pm$1.9 | **107.2$\pm$1.0** | **107.8$\pm$0.7** |
> |halfcheetah-e|  91.8$\pm$1.5  | 97.3$\pm$1.1 | 92.4$\pm$1.7 | 104.9$\pm$0.9 | **106.8$\pm$3.4** |104.0$\pm$0.8 | **105.2$\pm$0.7**|
> |hopper-m-e| 53.9$\pm$4.7 | 96.9$\pm$15.1 | **110.8$\pm$0.8** | 6.1$\pm$7.7 | 110.7$\pm$0.1 | 58.1$\pm$22.3 |**112.7$\pm$0.2** |
> |hopper-e | 107.7$\pm$9.7 | 106.5$\pm$9.1 | **110.5$\pm$0.4** | 1.1$\pm$0.5 | 110.1$\pm$0.1 | 77.0$\pm$43.9 | **112.8$\pm$0.2** |
> |walker2d-m-e | 90.1$\pm$13.2 | 109.1$\pm$0.2 | 110.1$\pm$0.3 | **116.7$\pm$1.9**  | 114.7$\pm$0.9 | 115.4$\pm$0.5 |**121.2$\pm$1.5**|
> |	walker2d-e | 108.7$\pm$0.2 | 109.3$\pm$0.1 | 108.3$\pm$0.3| 1.2$\pm$0.7 | **115.1$\pm$1.9** | 57.8$\pm$55.7 | **115.4 $\pm$ 0.5** |
>
>
> **Q2: The proposed algorithm requires computing an adversarial example $\hat s$, which is computationally expensive.**
>
> A2: The adversarial state sampling method is widely adopted in prior robust RL works [2][3] and offline RL [4]. Though sampling $\hat s$ can be computationally expensive, RORL significantly improves the robustness over random and adversarial observation perturbations, which is considerably helpful for real-world scenarios when the sensor error is unavoidable and potential malicious attacks need to be guarded against.
>
> For the implementation of maximizing Eq.(2) in RORL, we use zero-order optimization by first sampling several examples within $B_d(s, \epsilon)$ and choosing a perturbation to maximize Eq. (2), which has the advantage of simplicity and less computational cost. We remark that such simplification is only used in training. We perform the standard optimization process for adversarial attacks including 'min Q mixed order' and 'action diff mixed order'.
>
> We add the comparison of training time between RORL and other methods in Appendix C.5. RORL runs slightly faster than CQL and much faster than PBRL. RORL includes the OOD state-action sampling and additional adversarial training procedures, while we implement these procedures efficiently based on GPU operation and parallelism. We also list the average epoch time below for a quick review.
>
> |  | CQL |SAC-10| EDAC |PBRL|RORL|
> | :-----|  :----: | :----: | :----: | :----: |:----: |
> | Average Epoch Time (s)| 32.40 | 12.73 | 17.94 | 102.96| 29.56|
> | GPU Memory (GB)  | 1.4 | 1.3|   1.8  | 1.8| 2.1|
>
> [2] Shen Q, Li Y, Jiang H, et al. Deep reinforcement learning with robust and smooth policy. ICML, 2020.
>
> [3] Zhang H, Chen H, Xiao C, et al. Robust deep reinforcement learning against adversarial perturbations on state observations. NeurIPS, 2020.
>
> [4] Sinha S, Mandlekar A, Garg A. S4RL: Surprisingly simple self-supervision for offline reinforcement learning in robotics. CORL, 2022.

---

> > ### Author Response · Authors · 2022-08-02
> > **Response to Reviewer yLKL (part 2/2)**
> >
> > **Q3:  However, assuming that $\hat a$ is close to $a$, the proposed method has a similar effect as PBRL since the Q-functions are trained to have similar values for $s$ and $\hat s$. Could you please conduct an ablation study that only penalizes OOD actions like PBRL?**
> >
> > A3: In Figure 2, we demonstrate that the Q values can change dramatically even with small perturbation scale, implying that further conservatism on OOD states is needed.
> >
> > When the hyper-parameter OOD $\epsilon_{ood}=0$, RORL only penalizes in-distribution states and OOD actions similar to PBRL. In Appendix C.1 and Figure 8, we ablate RORL with $\epsilon_{ood}=0$. The results show that only penalize in-distribution states and OOD actions is not sufficient for robust value estimation, which still suffer from severe value overestimation. We listed the final performance and average value estimation below for a quick review.
> >
> > |  | RORL $\epsilon_{ood}=0$ | RORL |
> > | :-----| ----: | :----: |
> > | Normalized Score  | 65.3 $\pm$47.4  |102.4 $\pm$ 1.4  |
> > | Average Q values | 1.1 $\times$ $10^{13}$ $\pm$ 1.5 $\times 10^{13}$ | 318.1 $\pm$ 0.7 |
> >
> > **Q4: I think the paper is closely related to S4RL, which utilizes adversarial examples for smoothness. It would be better to provide the results of CQL + S4RL or PBRL + S4RL.**
> >
> > A4: Our method is different from S4RL since S4RL only performs different data augmentations to observation, while we enforce explicit smoothness for both the value function and the policy. Meanwhile, our method has theoretical guarantees in linear MDPs with tighter suboptimality gap. We conduct experiments of PBRL+S4RL in Appendix C.10 and Figure 16, and find S4RL only slightly improves the robustness of PBRL on the walker2d-medium-v2 task and has little impact on halfcheetah-medium-v2 task. We also report the 'weighted robust score' (i.e., the weighted areas under the performance curves in Figure 16, detailed definition is in Appendix C.4) on the halfcheetah-medium-v2 and walker2d-medium-v2 tasks below. The weighted robust score of PBRL+S4RL on the walker2d task is 1.1 points higher than PBRL, but still 8.9 points lower than RORL. More analysis about S4RL can be found in Appendix C.10.
> >
> > |     halfcheetah-m      |   random |   action diff |   action diff mixed order |   min Q |   min Q mixed order |   Average Score |
> > |:----------|:-----------------:|:----------------------:|:----------------------------------:|:----------------:|:----------------------------:|:----------------:|
> > | RORL      |             57.4 |                  44.5 |                              29.7 |            37   |                        17.7 |            37.2 |
> > | PBRL      |             56.7 |                  38.6 |                              25.9 |            30.2 |                        14.0   |            33.1 |
> > | PBRL+S4RL |             56.9 |                  38.8 |                              26.2 |            29.4 |                        13.9 |            33.0   |
> >
> > |     walker2d-m      |   random |   action diff |   action diff mixed order |   min Q |   min Q mixed order |   Average Score |
> > |:----------|:-----------------:|:----------------------:|:----------------------------------:|:----------------:|:----------------------------:|:----------------:|
> > | RORL      |             94.1 |                  89.1 |                              39.1 |            61.8 |                        26.7 |            62.2 |
> > | PBRL      |             93.2 |                  63.8 |                              33   |            50.8 |                        20.4 |            52.2 |
> > | PBRL+S4RL |             92.9 |                  62.9 |                              29.9 |            57.4 |                        23.2 |            53.3 |
> >
> > **Q5:  Please report the wall-clock time of the proposed method in the main text.**
> >
> > A5: We compare the training time between RORL and other methods in Appendix C.5. For each method, we measure the average epoch time and the GPU memory usage on the hopper-medium-v2 task.
> >
> > As shown in Table 8 (Appendix C.5), RORL runs slightly faster than CQL, mainly because CQL needs the OOD action sampling and the logsumexp approximation. For ensemble-based baselines, RORL is much faster than PBRL, requiring only 28.7$\\%$ of PBRL's epoch time. In RORL, we also include the OOD state-action sampling and additional adversarial training procedures, but we implemented these procedures efficiently using GPU operation and parallelism. Even so, RORL is still slower than SAC-10 and EDAC. But as demonstrated in our experiments, RORL enjoys significantly better robustness than EDAC and SAC-10 under different types of observation perturbation. As for the GPU memory consumption, RORL uses comparable memory to PBRL and EDAC, with only $16.7\\%$ more memory usage. The comparison of training time are shown in A2.

---

> > > ### Comment · Reviewer_yLKL · 2022-08-08
> > > **Response to paper7821 authors**
> > >
> > > Thanks for the detailed response. Since the authors have well addressed all my concerns, I decide to raise my score to weak accept.

---

> > > > ### Author Response · Authors · 2022-08-08
> > > > **Thanks for raising the score!**
> > > >
> > > > We would like to thank the reviewer for acknowledging that we have addressed his major concerns! We really appreciate the valuable comments and suggestions from the reviewer.

---

> ### Author Response · Authors · 2022-08-06
> **Looking forward to further feedback**
>
> Dear Reviewer yLKL,
>
> Since the author-reviewer discussion period has started for a few days, we will appreciate if you could check our response to your review comments soon. This way, if you have further questions and comments, we can still reply before the author-reviewer discussion period ends. If our response resolves your concerns, we kindly ask you to consider raising the rating of our work. Thank you very much for your time and efforts!
>
> Best,
>
> The authors

---

> > ### Author Response · Authors · 2022-08-07
> > **Dear Reviewer**
> >
> > Since there are only 2 days left in the author-reviewer discussion period, we would appreciate if you could check our reply to your comments soon. We believe that we have addressed the concerns mentioned in your review, please let us know if you have additional concerns.
> >
> > Thank you again for your time and efforts!

---

### Official Review · Reviewer_jjFr · 2022-07-15

**Rating:** 6
**Confidence:** 3
**Soundness:** 2 fair
**Presentation:** 3 good
**Contribution:** 2 fair

**Summary:**

This paper proposes a conservative smoothing technique by adding perturbation to the states to improve the robustness of learned policy on offline RL. Theoretically, they claim their work enjoys a tighter suboptimality bound in linear MDPs.

**Questions:**

More ablation study is necessary, especially for clarifying the contribution of the ensemble technique.

**Limitations:**

Yes.

**Strengths And Weaknesses:**

**Strengths:**
Prior offline RL methods mainly focus on the OOD actions instead of states. This paper targets the robustness issue for Q value via perturbing the states. This paper is well-written and easy to follow.

**Weaknesses:**
The algorithm employs the ensemble Q functions. It is well known that the ensemble technique can bring robustness to learning. However, it is unclear whether or not the ensemble technique brings the main contribution to the robustness. To this end, comparing single agent methods, such as CQL and BC, is unfair. Moreover, on the results of the experiments, the proposed RORL is also very close to the ensemble baseline EDAC.

---

> ### Author Response · Authors · 2022-08-02
> **Response to Reviewer jjFr (part 1/2)**
>
> Thanks for your thoughtful comments. We provide clarification to your questions as below. We appreciate it if you have any further questions or comments.
>
> Q1: it is unclear whether or not the ensemble technique brings the main contribution to the robustness...
>
> A1: Ensemble is an important component in RORL for uncertainty measurement, but simple ensemble does not bring a strong performance. As shown in the Table 1, SAC-10 performs even worse than CQL. RORL is implemented based on SAC-10 with only 10 ensemble Q networks, while EDAC uses 10-50 ensemble Q networks in their paper. We reproduce EDAC with only 10 ensemble Q networks in Table 1, and on average EDAC-10 obtains results 14 points lower than RORL. The results imply the significant improvement of RORL over previous ensemble baselines with the same number of ensemble Q networks. The average normalized scores over 15 tasks are also listed below for a quick review.
>
> |  | CQL | SAC-10 | EDAC-10 | RORL |
> | :-----| :----: | :----: | :----: | :----: |
> | Average Score | 70.2 | 50.8 | 71.2 | 85.3 |
>
> In addition, we also include an ablation on the number of ensemble Q networks in **Appendix C.6** and **Figure 13**. The robustness of RORL improves as the ensemble size $K$ increases. For $K = 6, 8, 10$, RORL has similar initial performance but $K = 10$ considerably outperforms others as the scale of attack increases. Therefore, we set $K = 10$ by default in our paper. The same number of ensemble Q networks are also used in prior offline RL work PBRL [1].
>
> [1] Bai C, Wang L, Yang Z, et al. Pessimistic Bootstrapping for Uncertainty-Driven Offline Reinforcement Learning[C]. ICLR 2022.

---

> > ### Author Response · Authors · 2022-08-07
> > **Response to Reviewer jjFr (part 2/2)**
> >
> > We also report the 'robust score' on the walker2d-medium-v2 task below, i.e., the areas under the performance curves in Figure 13 (detailed definition is in Appendix C.4). Note that RORL with 8 ensemble Q networks already outperforms EDAC and SAC-10 with 10 Q networks which only achieve average robust scores of 59.7 and 31.2 (in Table 6), respectively.
> >
> > |      |   Random |   Action diff |   Action diff mixed order |   Min Q |   Min Q mixed order |   Average Score |
> > |:-----|:-----------------:|:----------------------:|:----------------------------------:|:----------------:|:----------------------------:|:----------------:|
> > | K=10 |             94.1 |                  90.9 |                              56.9 |            71.0   |                        43.3 |            71.2 |
> > | K=8  |             93.2 |                  91.0   |                              56.1 |            53.5 |                        33.2 |            65.4 |
> > | K=6  |             78.3 |                  68.7 |                              40.8 |            39.8 |                        22.8 |            50.1 |
> > | K=4  |             34.8 |                  34.2 |                              24.1 |            14.2 |                        10.0   |            23.5 |

---

> > > ### Comment · Reviewer_jjFr · 2022-08-08
> > > **Response to paper7821 authors**
> > >
> > > Thanks for the response. As the authors have addressed my main concern, I decide to raise my score.

---

> > > > ### Author Response · Authors · 2022-08-08
> > > > **Thanks for raising the score!**
> > > >
> > > > We would like to thank the reviewer for acknowledging that we have addressed his major concerns! We really appreciate the valuable comments and suggestions from the reviewer.

---

> ### Author Response · Authors · 2022-08-06
> **Looking forward to further feedback**
>
> Dear Reviewer jjFr,
>
> Since the author-reviewer discussion period has started for a few days, we will appreciate if you could check our response to your review comments soon. This way, if you have further questions and comments, we can still reply before the author-reviewer discussion period ends. If our response resolves your concerns, we kindly ask you to consider raising the rating of our work. Thank you very much for your time and efforts!
>
> Best,
>
> The authors

---

> > ### Author Response · Authors · 2022-08-07
> > **Dear Reviewer**
> >
> > Since there are only 2 days  left in the author-reviewer discussion period,  we would appreciate if you could check our reply to your comments soon.  We believe that we have addressed the concerns mentioned in your review, please let us know if you have additional concerns.
> >
> > Thank you again for your time and efforts!

---

### Author Response · Authors · 2022-08-06
**General Response**

We thank all reviewers for their constructive comments and hope we addressed all of them in our response.

In summary, our response includes the following aspects:
1. **[Expert datasets]** Adding the performance on ‘expert’ datasets in Table 1.
2.  **[Computational Cost]**  Providing comparison and analysis on the training time and GPU memory usage in Appendix C.5.
3. **[Number of Ensemble]** Providing additional ablation results on the number of ensemble Q networks in Appendix C.6 and Figure 13.
4. **[Adroit Tasks]**  Providing additional results on the challenging Adroit benchmark in Appendix C.8 and Table 9.
5. **[Robust Measure]** Defining two robust measures according to the areas under the perturbation curves, and comparing the robust scores in Appendix C.4, Table 6 and Table 7.
6. **[IQL+Smooth]** Providing additional experimental results combining the smoothing techniques with IQL, a SOTA offline RL algorithm without ensemble Q networks, in Appendix C.7 and Figure 14.
7. **['Min Q' Attack with Different Q]** Providing additional experimental results on the adversarial attack with different Q functions in Appendix C.9 and Figure 15.
8. **[S4RL]**  Providing discussion and comparison with PBRL+S4RL in Appendix C.10 and Figure 16.
9. **[First-order Optimization]** Providing comparison with zeroth-order and first-order optimization for perturbation generation during training in Appendix C.11 and Figure 17.
10. **[AntMaze Tasks]** Providing additional experimental results combining the smoothing techniques with IQL on challenging AntMaze tasks in Appendix C.12 and Table 10.

We hope our response could address the reviewers' concerns. If you have any other questions, please post them. We are happy to have further discussions.

---

### Public Comment · ~Ezgi_Korkmaz2 · 2023-02-16
**Acknowledgement of Recent Studies**

This paper needs to refer recent studies [1,2,3] on adversarial deep reinforcement learning. More in particular, it has already been shown that certified adversarial training techniques are vulnerable to many different sets of attacks from perturbations that can transfer [2] to natural directions [1]. This study at a minimum should acknowledge and refer to these studies.

[1] Adversarial Robust Deep Reinforcement Learning Requires Redefining Robustness. AAAI Conference on Artificial Intelligence, 2023.

[2] Deep Reinforcement Learning Policies Learn Shared Adversarial Features Across MDPs. AAAI Conference on Artificial Intelligence, 2022.

[3] Investigating Vulnerabilities of Deep Neural Policies. Conference on Uncertainty in Artificial Intelligence (UAI), Proceedings of Machine Learning Research (PMLR), 2021.

---

### Meta-Review · Area_Chair_tMBn · 2022-08-24

**Recommendation:** Accept
**Confidence:** Less certain

**Metareview:**

All reviewers agree that the author's response has addressed their primary concerns. Reviewer frMM had two reservations that resulted in a borderline rating 1) concerns about how the adversarial samples were generated and 2) a request for evaluation on AntMaze. The author's followup response and further experiments address 1 and partially 2. It would be great to see RORL results on AntMaze in the final version.

Overall, the performance of RORL is competitive with state-of-the-art methods on Mujoco and Adroit tasks with fewer ensemble elements needed. The main benefit is on improved performance against adversarial attack, where RORL significantly improves over existing methods. I think the paper makes a nice contribution that the community will find valuable.

I encourage the authors to think carefully about how to integrate the additional experiments into the paper to resolve the questions raised by reviewers.




**Award:**

No

---

### Decision · Program_Chairs · 2022-09-14

Accept